# Mutation hotspots at CTCF binding sites coupled to chromosomal instability in gastrointestinal cancers

Yu Amanda Guo[1], Mei Mei Chang[1], Weitai Huang[1,2], Wen Fong Ooi[3], Manjie Xing[3,4], Patrick Tan[4,5] & Anders Jacobsen Skanderup [1]

Tissue-specific driver mutations in non-coding genomic regions remain undefined for most cancer types. Here, we unbiasedly analyze 212 gastric cancer (GC) whole genomes to identify recurrently mutated non-coding regions in GC. Applying comprehensive statistical approaches to accurately model background mutational processes, we observe significant enrichment of non-coding indels (insertions/deletions) in three gastric lineage-specific genes. We further identify 34 mutation hotspots, of which 11 overlap CTCF binding sites (CBSs). These CBS hotspots remain significant even after controlling for a genome-wide elevated mutation rate at CBSs. In 3 out of 4 tested CBS hotspots, mutations are nominally associated with expression change of neighboring genes. CBS hotspot mutations are enriched in tumors showing chromosomal instability, co-occur with neighboring chromosomal aberrations, and are common in gastric (25%) and colorectal (19%) tumors but rare in other cancer types. Mutational disruption of specific CBSs may thus represent a tissue-specific mechanism of tumorigenesis conserved across gastrointestinal cancers.

[1] Computational and Systems Biology, Agency for Science Technology and Research, Genome Institute of Singapore, 60 Biopolis Street, Singapore 138672, Singapore. [2] Graduate School of Integrative Sciences and Engineering, National University of Singapore, 5 Lower Kent Ridge Road, Singapore 117456, Singapore. [3] Cancer Therapeutics and Stratified Oncology, Agency for Science Technology and Research, Genome Institute of Singapore, 60 Biopolis Street, Singapore 138672, Singapore. [4] Cancer and Stem Cell Biology Program, Duke-NUS Medical School, 8 College Road, Singapore 169857, Singapore. [5] Cancer Science Institute of Singapore, National University of Singapore, 14 Medical Drive, Singapore 117599, Singapore. Correspondence and requests for materials should be addressed to P.T. (email: gmstanp@duke-nus.edu.sg) or to A.J.S. (email: skanderupamj@gis.a-star.edu.sg)

Non-coding DNA constitutes over 98% of the human genome and harbors numerous functional elements essential for regulating gene expression and maintaining chromosomal architecture[1]. Mutations at non-coding regions may drive cancer by dysregulating proto-oncogenes and tumor suppressor genes, as exemplified by recent studies demonstrating recurrent point mutations at the *TERT* promoter in multiple cancer types[2,3] and *TAL1* enhancer insertions in T-cell acute lymphoblastic leukemia[4]. While previous pan-cancer analyses of tumor genomes have nominated regulatory driver mutations[5,6], these studies have typically not been sufficiently powered to identify tissue-specific non-coding driver mutations, as hundreds of samples are usually needed to reliably identify driver mutations in individual cancer types[7]. Recently, the whole-genome mutational landscapes of breast[8], liver[9], and pancreatic[10] cancer tumors have been studied to identify cancer-specific non-coding drivers. However, the prevalence and impact of non-coding driver mutations is still unknown for most cancer types.

CTCF is a DNA-binding protein essential for the maintenance of genome architecture by mediating both short and long-range chromosomal contacts[11,12]. Together with the cohesin complex, CTCF organizes chromatin into large topologically associating domains (TADs), insulating the local chromosomal neighborhoods from adjacent regions. Disruption of CTCF binding can therefore lead to dysregulation of gene expression[11,12]. In cancer, CTCF binding is disrupted through various mechanisms, such as DNA copy number alterations spanning domain boundaries[13], microdeletions within CBSs[13], and hypermethylation of CBSs[14]. These alterations at CBSs may drive cancer progression by allowing ectopic expression of oncogenes. Notably, recent studies have reported a genome-wide elevated somatic mutation rate across CBSs in several cancer types[15–18]. This suggests that mutational and DNA repair processes may act differently at CBSs relative to other genomic regions, thereby resulting in an overall elevated mutational burden at such sites in cancer. However, no study to date has rigorously tested the hypothesis that even amidst this elevated mutational burden, positive selection may still act on specific CBSs to drive cancer in individual tumor types. To accurately identify such genomic sites under positive selection, statistical tests must take into account regional biases in the mutation burden.

Comprehensive genetic and molecular profiling have previously identified new molecular subtypes and genetic drivers of gastric adenocarcinoma[19–21]. Studies have also investigated the extent and impact of mutational signatures[22,23] and epigenetic dysregulation in gastric cancer (GC) genomes[24,25]. Yet, it is unknown to what extent mutations in specific non-coding elements may drive GC, a leading cause of global cancer mortality. Here, we performed uniform and accurate identification of somatic single nucleotide variants (SNVs) and insertions/deletions (indels) in 212 GC genomes using an ensemble mutation calling approach. We present a comprehensive statistical approach, incorporating both epigenetic and sequence covariates, to identify non-coding regions with significantly higher mutation burdens over background, indicating positive selection and a role in gastric tumorigenesis. Performing an unbiased genome-wide scan of focal mutation hotspots (~20 bp, as TF binding motifs are typically <20 bp), we detect 34 significant recurring non-coding hotspots—of these, 11 overlapped CBSs. We further characterize these sites by analyzing CBS specific mutation biases, gene expression of neighboring genes, chromosomal instability, and incidence of these mutations in other cancer types. Overall, our analyses nominate these CBS hotspots as candidate drivers of GC. Furthermore, our analysis suggests a general link between CBS mutations and chromosomal instability in gastrointestinal cancers.

## Results

**The mutation landscape of gastric adenocarcinoma.** We analyzed the whole-genome sequences of 212 gastric adenocarcinoma tumors and matched normal samples collated from four different sources (Supplementary Data 1; Methods). All samples were uniformly processed using an accurate somatic mutation calling pipeline (Supplementary Fig. 1a–b). Briefly, we trained a random forest classifier that predicts high confidence somatic mutation calls (SNVs and indels) by combining the outputs of four independent mutation callers. This approach achieved >85% accuracy on an independent test data set of curated somatic mutations[26]. We excluded 20 low-quality samples with less than 400 mutation calls from the discovery cohort (Supplementary Fig. 1c). In addition, we removed five samples with strong enrichment of C>A substitutions, a sign of oxidative damage during DNA preparation[27,28] (Supplementary Fig. 1c). Somatic mutations in coding (CDS) regions, immunoglobin loci, and poorly mappable regions were also removed from further analyses. After uniform processing, samples from the four cohorts showed comparable distributions of somatic mutation counts and similar mutation spectra (Fig. 1a and Supplementary Fig. 1a). The ICGC cohort had slightly fewer mutations per tumor, probably due to the lower sequencing depth of this cohort.

A previous study identified four molecular subtypes of gastric adenocarcinoma[19]: tumors that are EBV positive (EBV), tumors with high levels of microsatellite instability (MSI), tumors that exhibit copy number instability (CIN), and tumor that are genomically stable (GS). We investigated the correlations between somatic mutation rates of the four cancer subtypes and epigenetic profiles of gastric tissue obtained from the Roadmap Epigenomics project[29]. In general, somatic mutation rates were negatively correlated with regions of open chromatin (DNaseI hypersensitivity) and histone marks of active promoters (H3K4me3) and enhancers (H3K27ac) (Fig. 1b). The depletion of somatic mutations in regions of open chromatin is likely due to enhanced accessibility to the DNA repair machinery[30–32]. Notably, somatic mutations in the EBV subtype were less correlated with histone features and replication timing compared to the CIN and GS subtypes, suggesting that additional mutational biases may exist in EBV infected tumors.

Tumors belonging to the MSI subtype displayed strikingly different associations between epigenetic features and mutation patterns. We observed little association between mutation rate and open chromatin marks or replication timing in MSI tumors. This is likely because mismatch repair (MMR) deficient MSI tumors have lost MMR-coupled enhanced repair efficiency at early replicating open-chromatin regions[31]. In addition, we found that MSI mutation profiles showed a strong positive association with heterochromatin (H3K9me3) and repressive domains (H3K27me3) (Fig. 1b). This is in contrast with a previous study by Supek et al. reporting that mutations generated after MMR inactivation are no longer enriched in heterochromatin regions, arguing that genome-wide regional mutation rate variation is mostly a result of MMR[31]. Instead, our data suggests that, in addition to MMR, other repair or mutational processes may further contribute to variation of the GC mutation landscape. Principal component analysis (PCA) on the correlation matrix between the mutation profiles of individual tumors and the epigenetic covariates also revealed MSI tumors as a distinct cluster (Fig. 1c). Accordingly, we removed the small number of MSI tumors ($N = 19$) from the discovery cohort to ensure all tumors had similar mutational biases.

**Statistical framework for mutational hotspot identification.** To identify positive selection in cancer genomes, it is essential to

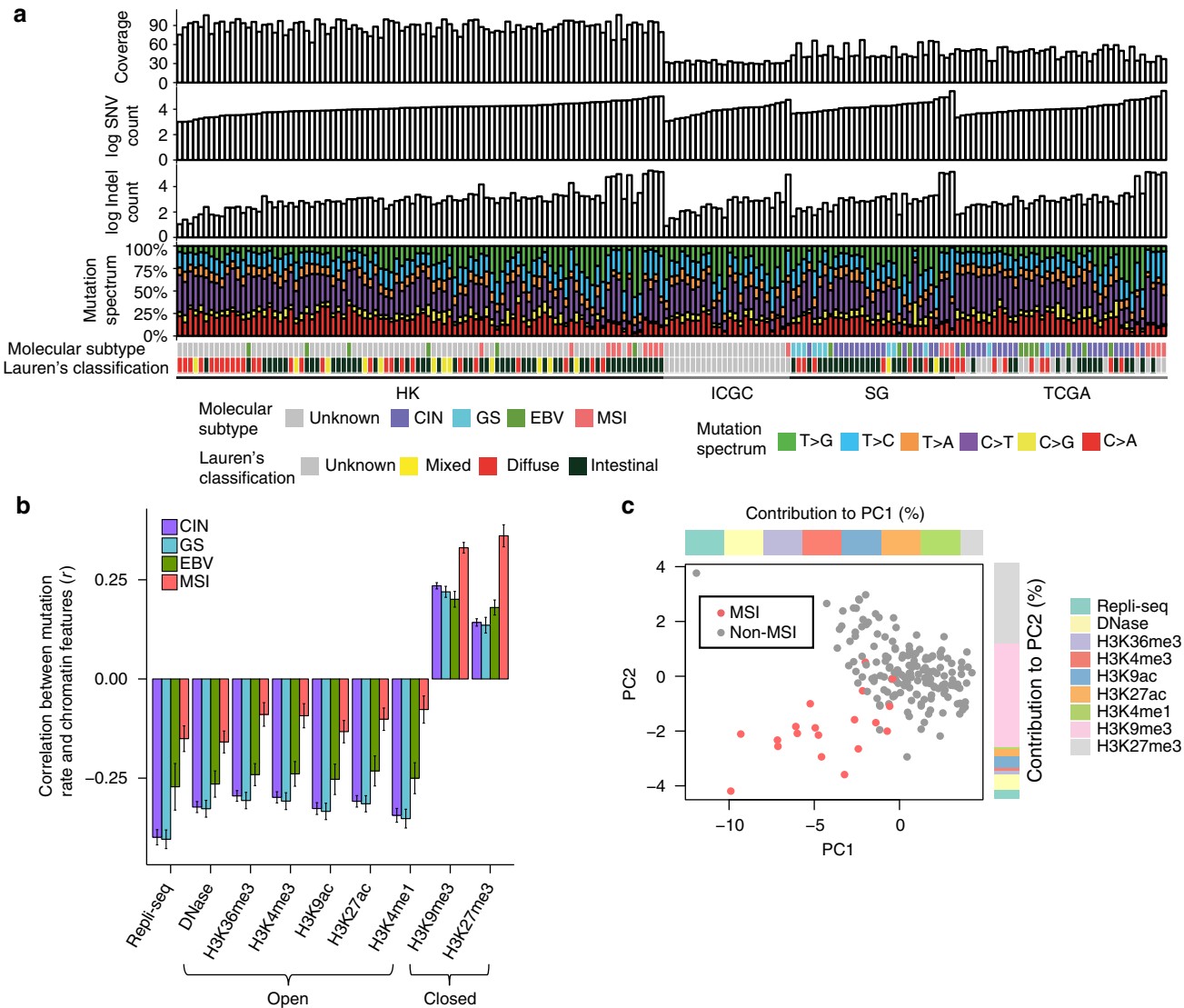

**Fig. 1** Summary of the data. **a** Gastric tumor samples grouped by cohort and ordered by SNV count within each cohort. The panels show coverage, SNV count, indel count, mutation spectrum, molecular subtype, and Lauren's classification of each sample. **b** Correlations between epigenetic features and somatic mutation rates in different tumor subtypes (error bars represent s.e.m of the correlation coefficient). **c** Principle component analysis of contributions of epigenetic features to the variance in the mutation rate of individual tumors. Colored stacked bars show the contribution of individual epigenetic features to the first two principal components

build an accurate background mutation rate model that corrects for covariates (features) that impact regional mutation rate variation, such as local sequence context and chromatin profiles. We considered a range of genetic and epigenetic features that could be correlated with GC somatic mutation rates. The features included 33 general and 36 gastric-specific chromatin features, 133 transcription factor binding profiles, and DNA replication timing profiles (Supplementary Data 2; Methods). To model the effect of local sequence context on mutation rate, previous studies have considered the single or tri-nucleotide sequence context of each mutation[5,6,8,9]. However, as mutation rates may also be influenced by wider sequence contexts[33], we thus used an expanded sequence context model that considers the effects of tri-nucleotide (1 bp flanks) and penta-nucleotide (2 bp flanks) contexts on the mutation probability of each base. Least absolute shrinkage and selection operator (LASSO) logistic regression was used to identify the most predictive epigenetic and sequence context features (Supplementary Fig. 2). We used these features to estimate sample-specific background mutation probabilities, and

to identify individual focal regions (21 bp) exhibiting mutational recurrence across samples beyond chance expectation (Fig. 2a; Methods). Overlapping significantly mutated regions were merged to obtain a list of unique hotspots.

**Recurrent indels in gastric lineage-specific genes.** We used this statistical framework to identify somatic mutation hotspots (both indels and point mutations) across the non-coding genome (Fig. 2b and Fig. 3a). The top indel hotspot was located ~100 kbp upstream of the *AFDN* gene, which is frequently translocated in leukemia and down-regulated in multiple cancer types[34–36]. The effect of hotspot mutations on *AFDN* expression could not be tested, as we lacked paired tumor expression data for the mutated samples. The second most significant indel hotspot was located in an intron of the *PGC* gene, which encodes the precursor of gastric proteinase pepsinogen (Supplementary Table 1). *PGC* is expressed at 11940 TPM in the stomach, 39 TPM in the lung, and ≤2 TPM in all other tissues in the Genotype-Tissue Expression (GTEx) project[37,38]. Interestingly, a recent study reported that

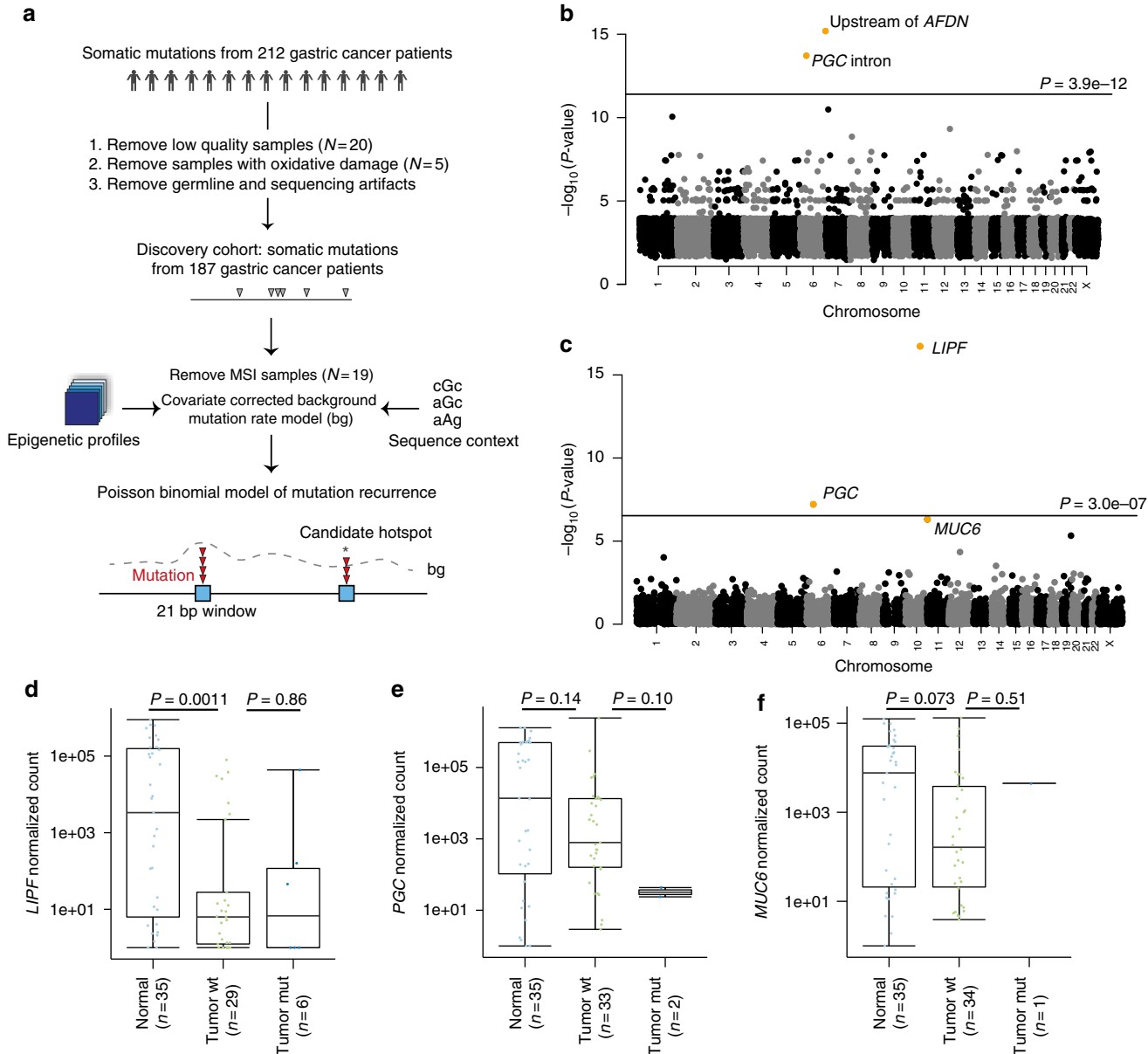

**Fig. 2** Genome-wide analysis of non-coding indel recurrence. **a** Workflow of the method to detect recurrently mutated non-coding regions. **b** Genome-wide negative log P-values of indel recurrence of 21 bp regions with at least one indel. The horizontal line marks the Bonferroni adjusted P-value of 0.01. **c** Negative log P-value of indel recurrence in merged non-coding regions of each gene. The top three significantly mutated genes are highlighted. The horizontal line marks the Bonferroni adjusted P-value of 0.01. **d–f** Gene expression of LIPF (**d**), PGC (**e**), and MUC6 (**f**) in normal gastric samples, tumors wildtype for the gene of interest, and tumors with non-coding indels in the gene of interest

*LIPF*, a lineage-specific gastric lipase, has broad enrichment of indels in GC[39]. Hypothesizing that other lineage-specific genes could show similar patterns of indel enrichment, we performed a gene-based recurrence analysis to identify all genes with broad enrichment of indels in their non-coding regions (combining promoter, untranslated, and intronic regions for each gene; Methods). Interestingly, the top three genes in this analysis were all lineage-specific genes highly expressed in stomach tissue: *LIPF, PGC,* and *MUC6* (Fig. 2c; Supplementary Table 2). *MUC6* encodes a mucin glycoprotein that is a major constituent of the gut mucosa, and is expressed at 133 TPM in stomach tissue, 38 TPM in the pancreas, and ≤2 TPM in all other tissues in GTEx[37,38]. However, consistent with the previous report[39], non-coding indels in these three recurrently mutated lineage-specific genes were not associated with expression change (Fig. 2d–f).

**Mutation hotspots enriched at CBSs in GC.** We then performed a genome-wide analysis of SNVs in non-coding regions and identified 34 significant mutation hotspots (Bonferroni adjusted P-value < 0.01; Fig. 3a; Supplementary Table 3). These hotspots were enriched in conserved sequences and TF binding regions, suggesting that many hotspot mutations may disrupt functional elements (Fig. 3b). Strikingly, of the 34 mutation hotspots, 11 were located in CBSs (Fig. 3a, c). The majority of mutations at CBS hotspots occurred in CIN tumors (71%, P = 0.012 by two-sided Fisher's exact test), which is the most common GC subtype, accounting for ~50% of all GC cases (Fig. 3c). The remaining 23 non-CBS hotspots often overlapped gene regions, but never co-located with TF binding regions. Furthermore, we observed a depletion of somatic mutations at gastric-specific TFBSs among the non-hotspot mutations (Fig. 3b). Overall, gastric tissue TFBSs

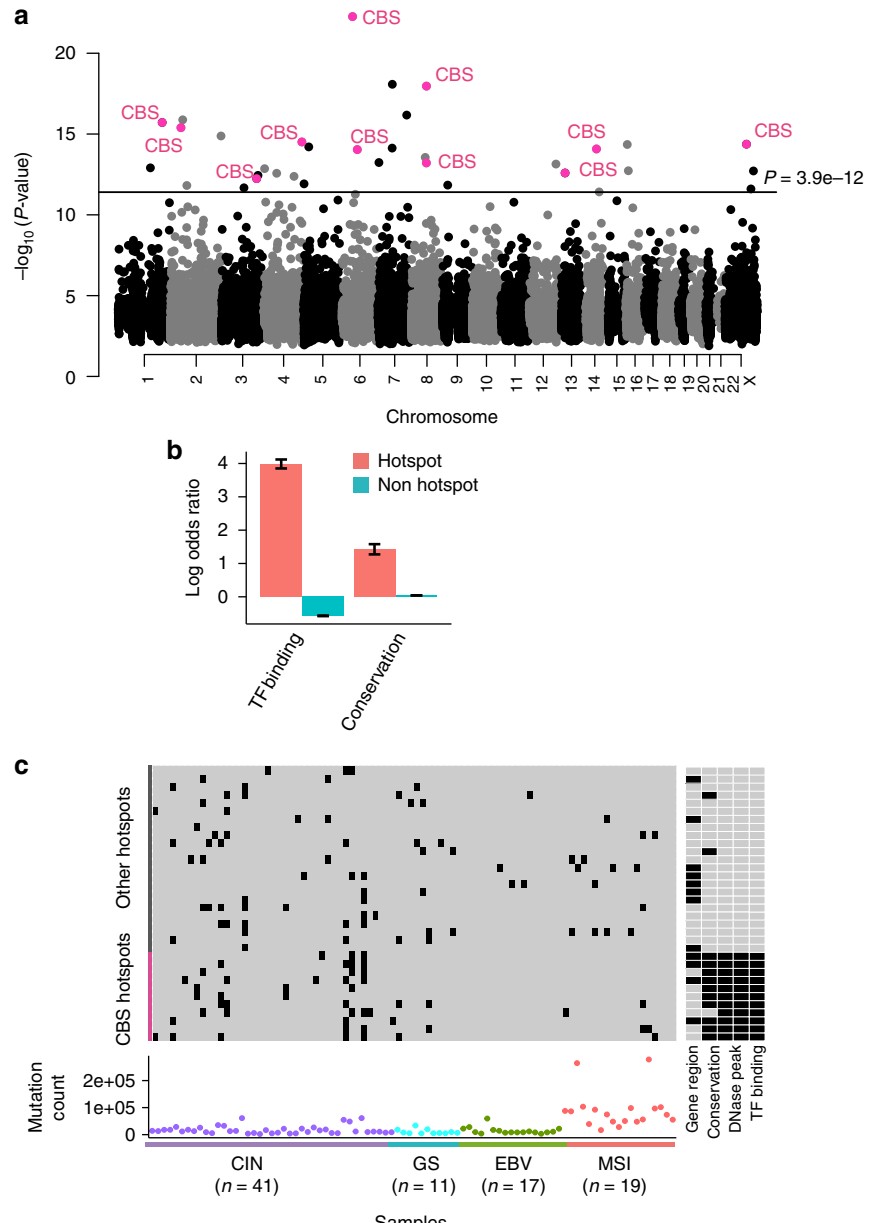

**Fig. 3** Genome-wide analysis of non-coding SNV hotspots. **a** The negative log *P*-values of SNV recurrence for all 21 bp regions genome-wide, only regions with at least three mutations are displayed. Significantly mutated hotspots overlapping CBSs are highlighted. The horizontal line marks the Bonferroni adjusted *P*-value of 0.01. **b** Log odds ratio of the enrichment of hotspot mutations and non-hotspot mutations in transcription factor binding regions and conserved regions. Error bars indicate the s.e.m of the log odds ratio. **c** Gastric cancer samples sorted by molecular subtype, with each row representing a significant mutation hotspot. Mutated samples are highlighted in black in the matrix. The mutation load of each sample is shown in the bottom panel. The right panel annotates the location of each hotspot with respect to annotated functional regions

comprises about 1% of the genome, but only 0.58% of the non-hotspot mutations were located in these regions. A similar depletion of mutations was observed for constitutive TFBSs (Supplementary Fig. 3). This is striking, as two recent studies have found that somatic mutation rates are elevated at transcription factor binding sites (TFBSs), and that this higher overall mutation load at TFBSs may be explained by reduced accessibility to nucleotide-excision repair (NER) enzymes at these sites[16,17]. This phenomenon is primarily observed in melanoma and lung adenocarcinoma where NER plays an important role in repairing carcinogen induced DNA lesions[16]. In contrast, our finding demonstrates that NER and TF occupancy is not a cause of regional mutational bias in GC.

To test if the 21 bp window size was adequate to capture most mutation hotspots, we repeated the hotspot analysis using larger 41 bp windows. In general, the rankings of the hotspots remained stable (Supplementary Fig. 4). 17/34 hotspots remained significant and only two additional hotspots were identified (*P* < 0.01, Bonferroni correction).

**Differential CBS mutation load across GC subtypes.** Despite the general depletion of somatic mutations at TFBSs in gastrointestinal tumors, several studies have reported an increased mutation rate specifically at CBSs in gastrointestinal tumors[15,18,40]. Indeed, when we examined all CBS across the genome, we found a 3-fold increased mutation rate at CBSs (11

mutations/Mb) compared to their 1 Kb flanking regions (3.6 mutations/Mb). In addition, the mutation frequencies at CBSs were very different among tumors of different molecular subtypes. The somatic mutation rate was 7.1 and 4.7-fold higher at CBSs compared to flanking regions in CIN and GS tumors, respectively (Fig. 4a–e). There was no enrichment of somatic mutations at CBSs in MSI tumors, likely due to impaired DNA MMR. Surprisingly, EBV tumors, which are not MMR-deficient, only had a modest 1.7-fold increase in mutation load at CBSs. The enrichment of somatic mutations at CBSs is therefore unlikely the result of differential DNA repair alone.

Consistent with findings by Katainen et al. in colorectal cancer[15], we found that somatic mutations at CTCF motifs, including the CBS hotspot mutations, were predominately A. T>C.G and A.T>G.C substitutions (Fig. 4f), suggesting that hotspot mutations are generated by the same mutational process as other CBS mutations. The mutation pattern at CBS hotspots was overall similar to that of all CBSs. However, while a conserved base at position 9 of the 19 bp CTCF binding motif was the most commonly mutated position at CBSs in general, the CBS hotspot mutations had the highest enrichment in the 4 bp sequence flanking the 5′ end of the CTCF motif. Furthermore, C>T changes, which are relatively common among all CBS mutations are much rarer among the CBS hotspot mutations (Fisher's exact test P-value = $4.4 \times 10^{-7}$). These differences could indicate a functional difference between CBS hotspot and non-hotspot mutations.

**Hotspots remain significant with a CBS-specific model.** To explicitly test if the CBS hotspots could be explained by the genome-wide elevated mutation rate at CBSs, we constructed a CBS-specific background mutation model. Since CBS mutation rates varied across tumor subtypes, this model further included the tumor subtype as a covariate. Also, since CBSs located at chromatin loop boundaries have higher somatic mutation burden than non-boundary CBSs[15,18], the CBS-specific background model differentiated between CBSs inside and outside chromatin loop boundaries. CTCF loop domains have not been profiled in gastric tissue but tend to be cell-type invariant[41,42]. We therefore used a constitutive set of CTCF domains shared across three cell lines (CM12878, Jurkat, and K562) to define CTCF loop boundaries[13,43]. In addition, since the mutation spectrum at CBSs is distinct from the overall genomic mutation spectrum, we performed LASSO logistic regression to identify sequence context features correlated with the somatic mutation rate at CBSs. To identify other mutational processes that might be associated with the occurrence of CBS mutations, we calculated the correlation between the proportion of CBS mutations in each tumor and the percentage contribution of each COSMIC mutation signature to each tumor[44]. While CBS mutations are known to be positively associated with signature 17[15], we found that CBS mutations were also strongly negatively associated with COSMIC mutation signature 1, an age related signature (Pearson correlation = −0.41; Supplementary Fig. 5). Therefore, we added the percentage contributions of mutation signatures 1 and 17 in each individual as covariates. Finally, this model also corrected for replication timing and local mutation rate. With this model, 9/11 CBS hotspots remained significant at the Bonferroni corrected significance threshold of 0.01 and the other two were borderline with adjusted P-values of 0.025 and 0.086 (Fig. 4g). Furthermore, seven additional CBSs became significant with the restricted hypothesis testing (Supplementary Table 4; Supplementary Figs. 6, 7). Mutations at these specific sites can therefore not be explained by a genome-wide elevated mutation rate at CBS, indicating that mutations at these focal sites are may be positively selected in gastric tumors.

**CBS hotspot mutations associated with gene expression changes.** We next examined the possibility that the CBS hotspots were associated with changes in expression of nearby genes. We restricted the analysis to the four CBS hotspots that had at least three mutated samples with gene expression data in the TCGA cohort (N = 35 samples). We validated the results using expression data from the SG cohort (N = 14 samples). Since the chromatin structure is generally cell-type invariant[41,42] and there is no published Hi–C data from gastric tissue, we used the Hi–C data from IMR90 cells published by Dixon et al.[41] to examine the 3D chromatin structure around each hotspot (Supplementary Figs. 8–10). We identified the flanking TAD boundary nearest to each hotspot, and tested the association between the mutation status of each hotspot and the expression of genes within the two adjacent TADs. We found genes with nominally altered expression for three of the four hotspots (Fig. 5), two of them remain significant after correcting for multiple testing in each region.

The first hotspot we identified is located in a CBS on chromosome 6 and has mutations in 12 samples (Fig. 5a–c). The expression of two neighboring genes, CENPQ and MUT, ~1 Mb upstream of this hotspot was significantly elevated in the mutated samples (P = 0.007 and 0.0021 respectively, Benjamini–Hochberg adjusted P = 0.026 and 0.042 respectively, two-sided Wilcoxon rank-sum test; Fig. 5a–c). A similar trend of CENPQ expression was observed using the expression data from the SG cohort (Supplementary Fig. 11a). CENPQ is a subunit of a centromeric complex, and is involved in mitotic progression and chromosomal segregation[45]. Interestingly, the tumor with the highest expression of CENPQ was mutated at the highly conserved position 9 of the CTCF motif, while the other two tumors were mutated at position 2 of the CTCF motif. This indicates that different mutations in the same hotspot may have different disruptive potentials. However, a formal evaluation of such effects requires a larger set of tumor samples with both CBS mutations and RNA-seq data available.

The next hotspot we tested is located on chromosome 6 with 9 mutated samples. Tumors with mutations at this hotspot had significantly lower expression of the KCNQ5 gene (Wilcoxon P = 0.0059, adjusted P = 0.047), located ~200 kb downstream of the hotspot (Fig. 5d–f). A similar trend in KCNQ5 expression was observed using the expression data from the SG cohort (Supplementary Fig. 11b). A recent study by Umer et al. found the same mutation hotspot by analyzing motif-breaking mutations[40]. Using an electrophoretic mobility shift assay, Umer et al. confirmed that the chr6:73,122,103 A>G mutation disrupts CTCF binding. In addition, it has been reported that CTCF is involved in the spatial organization of the KCNQ5 locus, and knockdown of CTCF downregulates KCNQ5 expression[46].

At the third hotspot located on chromosome 13, mutated tumors had on average a 3-fold decrease in SPG20 expression (Wilcoxon P = 0.045, adjusted P = 0.65; Fig. 5g–i). However, only three tumors with expression data were mutated at this hotspot, and the expression change was not significant after correcting for multiple testing. A larger sample size is needed to evaluate if this is a spurious or true correlation. A similar trend in SPG20 expression is observed using the expression data from the SG cohort (Supplementary Fig. 11c). SPG20 is involved in epidermal growth factor receptor trafficking[47] and was previously found to be significantly mutated in the exome of esopheagal cancer[48].

In all three cases, we confirmed that the expression changes of these genes were significant after correcting for variation in DNA copy numbers and tumor purity between samples (Supplementary Fig. 12). We would expect mutations at these hotspots to be associated with expression change of the altered allele. However, allele-specific expression analysis is challenging to evaluate across

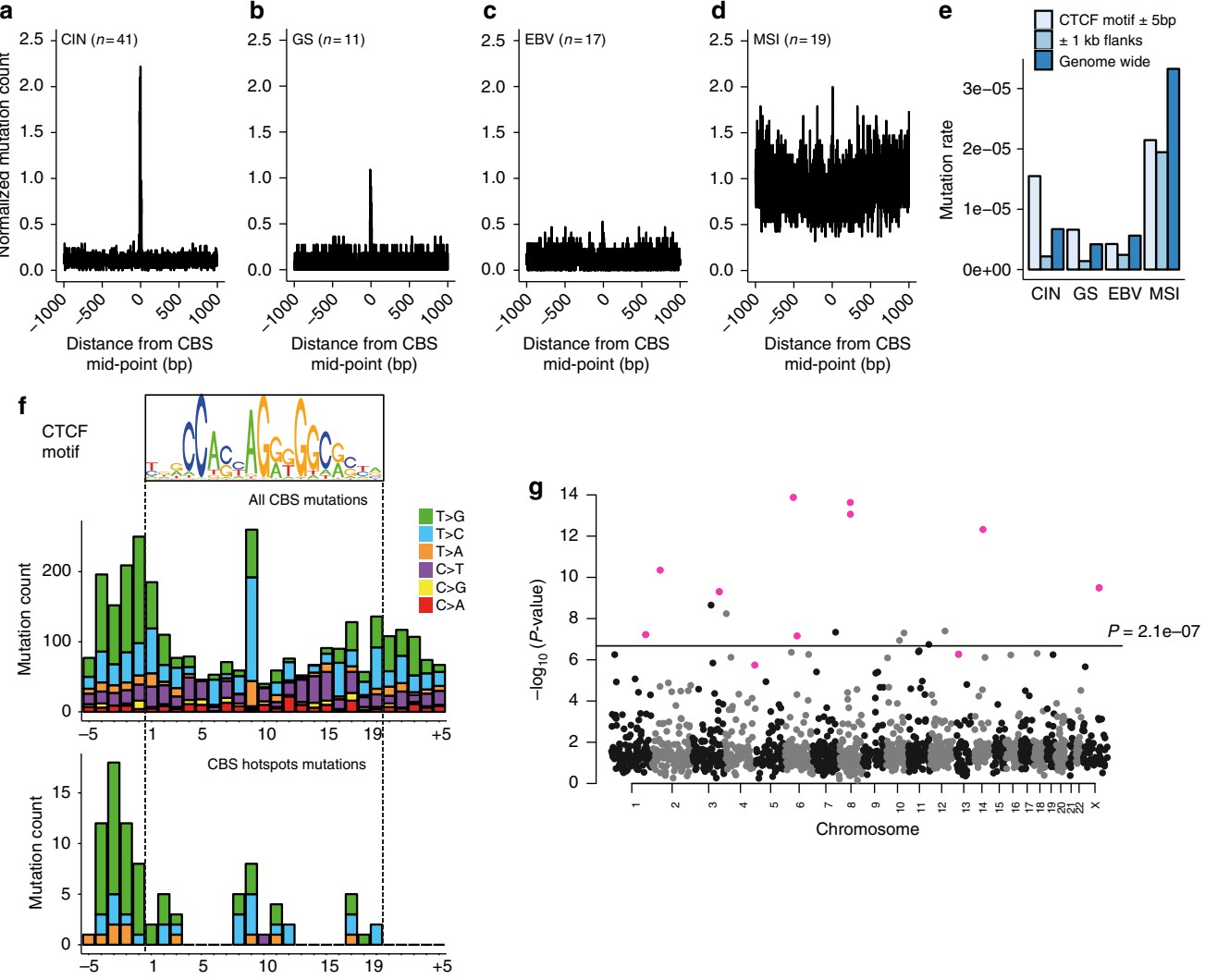

**Fig. 4** Analysis of CBS mutations in different gastric cancer subtypes. **a–d** Mutation count per tumor around CBSs in the four gastric cancer subtypes. **e** Elevated mutation rates at CBSs compared to flanking regions. **f** Somatic substitution patterns within CTCF motifs for hotspot mutations and all mutations, respectively. **g** The negative log P-values of mutation recurrence of all CBSs evaluated with a CBS-specific background model. CBS hotspots identified in Fig. 3a are highlighted in magenta. The horizontal line marks the Bonferroni adjusted P-value of 0.01

distant sites. Indeed, for all three candidate CBS hotspots, we could not unambiguously resolve the two alleles because the maximum inter-SNP distance between hotspot and candidate target gene were >5 kb. As CBSs are essential in maintaining the chromosomal architecture, it is likely that these CBS hotspot mutations cause altered expression of nearby cancer driver genes by disrupting the local chromosomal organization. Indeed, using the set of constitutive CTCF-CTCF loops, we observed chromatin contacts between the *KCNQ5* and *SPG20* loci and their corresponding CBS hotspots (Supplementary Figs. 9, 10). Interestingly, the three genes were also differentially expressed in GC tumors compared to normal gastric tissue. *CENPQ* expression was upregulated in tumors (Wilcoxon $P = 0.0028$; Fig. 5c), while both *KCNQ5* and *SPG20* expression was downregulated in tumors compared to normal gastric samples (Wilcoxon $P = 3.2 \times 10^{-7}$ and 0.00082 respectively; Fig. 5f, i). Therefore, it is plausible that the expression of these three genes could be altered in GC through additional mechanisms. Indeed, *KCNQ5* and *SPG20* were found to be downregulated in colorectal cancer compared to the normal mucosa due to promoter hypermethylation[49–51]. These results further support the contributions of these genes to GC tumorigenesis.

Many of the hotspot mutations were located in the 5′ flanks of the consensus CTCF motif (Fig. 4f). Previous studies have found increased conservation in the flanking sequences of weak CTCF and REST binding sites, suggesting that the sequence context is important for TF binding at these sites[52,53]. We examined the evolutionary conservation[54] at the CTCF binding motifs and their flanking sequences. In general, the 5′ flanks of the CTCF motifs are not conserved (Supplementary Fig. 13a). However, in the hotspot upstream of *CENPQ*, the mutation cluster in the 5′ flank co-occurred with conserved bases (Supplementary Fig. 13b). In addition, we found another CBS hotspot with nine 5′-flank mutations that coincided with a highly conserved base (Supplementary Fig. 13c). Such hotspot mutations, affecting conserved 5′ flanks of CTCF motifs, could disrupt context-specific binding of CTCF.

We also examined the possibility that mutations in the flanking regions of CTCF motifs create or disrupt binding motifs of other TFs. We used DeepBind[55] to predict the binding scores of wildtype and mutated sequences for 472 transcription factors. However, we only found mutations at three CBS sites with predicted change in TF binding (Supplementary Table 5). Lastly, it is also possible that some mutations at CBS flanks are passenger mutations arising due to the overall elevated mutation rates at

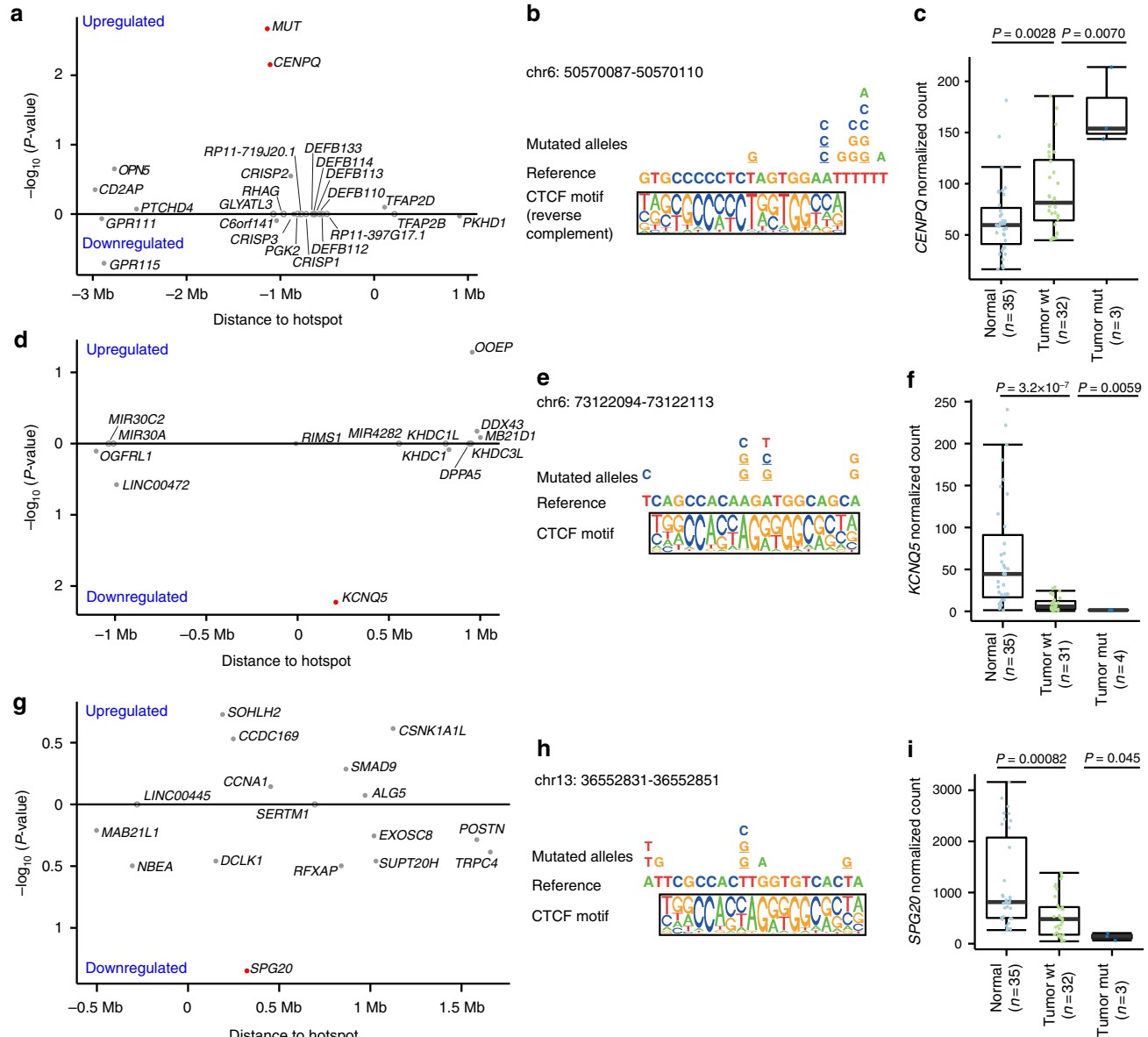

**Fig. 5** Association of CBS hotspot mutations and cis-gene expression. **a, d, g** Association between mutation status of the CBS hotspot and expression levels of neighboring genes (two-sided Wilcoxon rank-sum test). Upregulated genes are shown above the x-axis, and downregulated genes are shown below the x-axis. Non-expressed genes are shown with empty circles on the x-axis (normalized count < 10 in all samples). **b, e, h** The reference sequence and mutated alleles at the three CBS hotspots. The mutations in tumors with expression data are underlined. **c, f, i** The gene expression of *CENPQ* (**c**), *KCNQ5* (**f**), and *SPG20* (**i**) in normal gastric tissue, and tumors with and without mutations at the corresponding CBS hotspot. P-values were adjusted using the Benjamini–Hochberg method

CBSs. While our model identifies individual CBS regions with overall mutation enrichment, it does not allow us to distinguish between passenger and driver mutations within such regions.

**CBS hotspots are often mutated in gastrointestinal cancers.** Taken collectively, 25% of all gastric tumors are mutated in at least one of the 11 CBS hotspots, representing the second most mutated functional region in GC after *TP53* (50% of gastric tumors). To study if these hotspots could also play a role in other cancer types, we examined the recurrence of these 11 hotspots in 826 non-hypermutated tumors of 18 other cancer types[5] (Fig. 6 and Supplementary Fig. 15). Strikingly, we found that 19% of colorectal cancer tumors were mutated at one or more of the CBS hotspots (Fig. 6a). Since colorectal cancer have pathological and molecular similarities to GC[56], the CBS hotspot mutations may

drive cancer progression in colorectal cancer through similar mechanisms as in GC. The CBS hotspots were mutated at lower frequencies in breast cancer, liver cancer, lung cancer, pancreas cancer, and lymphoma. Interestingly, while melanoma and bladder carcinoma also have high genome-wide mutation rates at CBS, none of the CBS hotspots were mutated in these two cancer types. Similarly, we found that mutations at all CBS hotspots had previously been reported in COSMIC[57] or other genome-wide studies of gastrointestinal tumors[15,40] (Supplementary Table 6). This suggests that the CBS hotspot mutations are generated and act in a cancer-specific manner.

**CBS mutations are associated with chromosomal instability.** Enrichment of CBS mutations was highest in CIN tumors, which are characterized by increased chromosomal aneuploidy. This

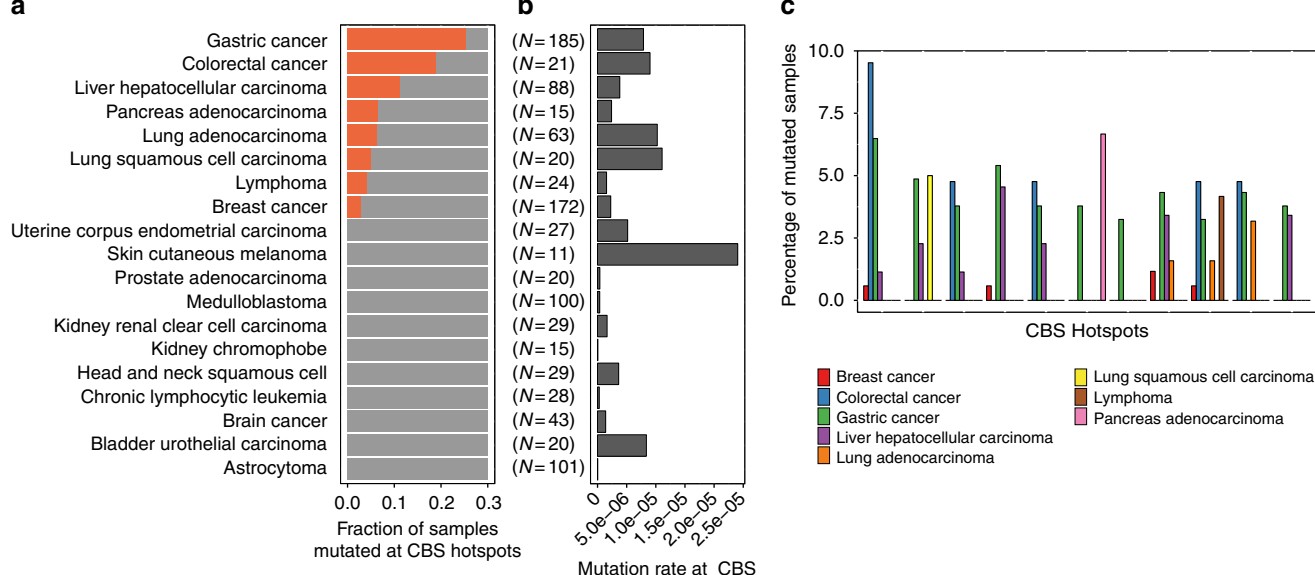

**Fig. 6** Pan-cancer analysis of mutation recurrence at the 11 CBS mutation hotspots. **a** Fraction of samples with mutation in at least one of the CBS hotspots in different cancer types. **b** Mutation rate of CBSs in different cancer types. **c** Mutation recurrence of individual CBS hotspots in different cancer types

prompted us to examine if mutations at CBSs in CIN tumors were correlated with somatic copy number alteration (SCNA) breakpoints. Strikingly, the distance between a CBS hotspot and its nearest SCNA breakpoint was significantly shorter in mutated than non-mutated tumors ($P = 0.0018$, two-sided Wilcoxon rank-sum test; Fig. 7a). In contrast, non-CBS mutation hotspots showed no such association ($P = 0.53$). The median distance between CBS hotspot mutations and its nearest SCNA breakpoint in the same sample was ~1 Mbp, notably shorter than the ~2 Mbp distance for non-CBS hotspots (Fig. 7a). To study whether this correlation between CBS mutations and SCNA breakpoints was specific to the CBS hotspots, we extended the analysis to all CBSs. Interestingly, we found that CBS mutations were correlated with occurrence of nearby SCNA breakpoints in the same samples, especially for mutations affecting CBSs at loop boundaries (Wilcoxon $P = 5.7 \times 10^{-16}$; Fig. 7b). Conversely, when we grouped 1 Mb windows of the genome according to SCNA breakpoint density, we found that the normalized CBS mutation rate was positively associated with SCNA breakpoint density (Fig. 7c–d). Overall, these results highlight a link between regional chromosomal instability and mutations at both CBS hotspots and boundary CBSs in general.

As the CBS mutation rate was also elevated in GS tumors (Fig. 4b), we investigated if there was a similar association between CBS mutations and SCNA in GS tumors. Although we found that mutated CBSs also tended to be closer to SCNA breakpoints compared to the non-mutated CBSs in GS tumors, the difference was not statistically significant (Supplementary Fig. 14), and the relative difference was greater in CIN (2.17-fold difference in distance to nearest breakpoint) compared to GS (1.58-fold difference) tumors. This may indicate that the coupling of CBS mutations and nearby chromosomal instability is a process that is specific to, or exacerbated in, the CIN tumors.

## Discussion

In this study, we performed a comprehensive and unbiased analysis of non-coding SNVs and indels in 212 GC genomes, the largest studied cohort thus far. In addition to a previously identified indel enrichment at *LIPF*[39], our analysis identified two other gastric lineage-specific genes with broad enrichment of

non-coding indels (*PGC* and *MUC6*). Our results show that the accumulation of indels occur in multiple lineage specific genes in GC. Yet, indels at these three genes were not associated with change in gene expression. The functional consequences of these indels are therefore still unclear. Strikingly, genome-wide analysis of somatic SNVs revealed 34 significant mutation hotspots (Bonferroni adjusted *P*-value < 0.01) that were disproportionately enriched in CBSs. Katainen et al. previously reported an increased mutation load at CBSs in colorectal cancer[15], and a subsequent study by Kaiser et al. confirmed the general hypermutation at CBSs in 11 cancer types[18]. Both studies generally discounted CBS mutations as passengers, yet they did not rigorously explore the hypothesis that a subset of these mutated CBSs may be undergoing positive selection within individual cancer types. Indeed, a recent study on motif-breaking mutations identified a recurrent CBS mutation that disrupts CTCF binding[40], confirming the motif-breaking potential of CBS mutations. Here, we used a large cohort of GC genomes in combination with rigorous statistics, to show that mutation rates at 11 specific CBSs are unexpectedly high and cannot alone be explained by a genome-wide elevated mutation burden at CBS, indicating positive selection at these sites. Out of the four CBS hotspots we examined, three of them were associated with nominally significant expression changes of neighboring genes (*CENPQ*, *KCNQ5*, and *SPG20*), and these associations were validated in an independent tumor cohort. Furthermore, it is possible that mutations at these CBS hotspots also have long-range or spatio-temporal regulatory effects on gene expression that are not captured by bulk tumor transcriptome profiling. Overall, our analyses nominate these CBS hotspots as potential drivers in GC, and support the hypothesis that driver mutations may arise as a by-product of the increased mutation load at CBSs followed by positive selection at specific CBSs. This is comparable to a model of genomic rearrangement hotspots in breast cancer, where rearrangements initially arise from defective homologous-recombination-repair and those affecting cancer risk loci are subsequently positively selected, forming rearrangement hotspots[58].

We found that gastric tumors of the genomic instable subtype (CIN) exhibited the highest mutation rate at CBSs compared to tumors of the other GC subtypes. Furthermore, CBS mutations were associated with the occurrence of nearby chromosomal

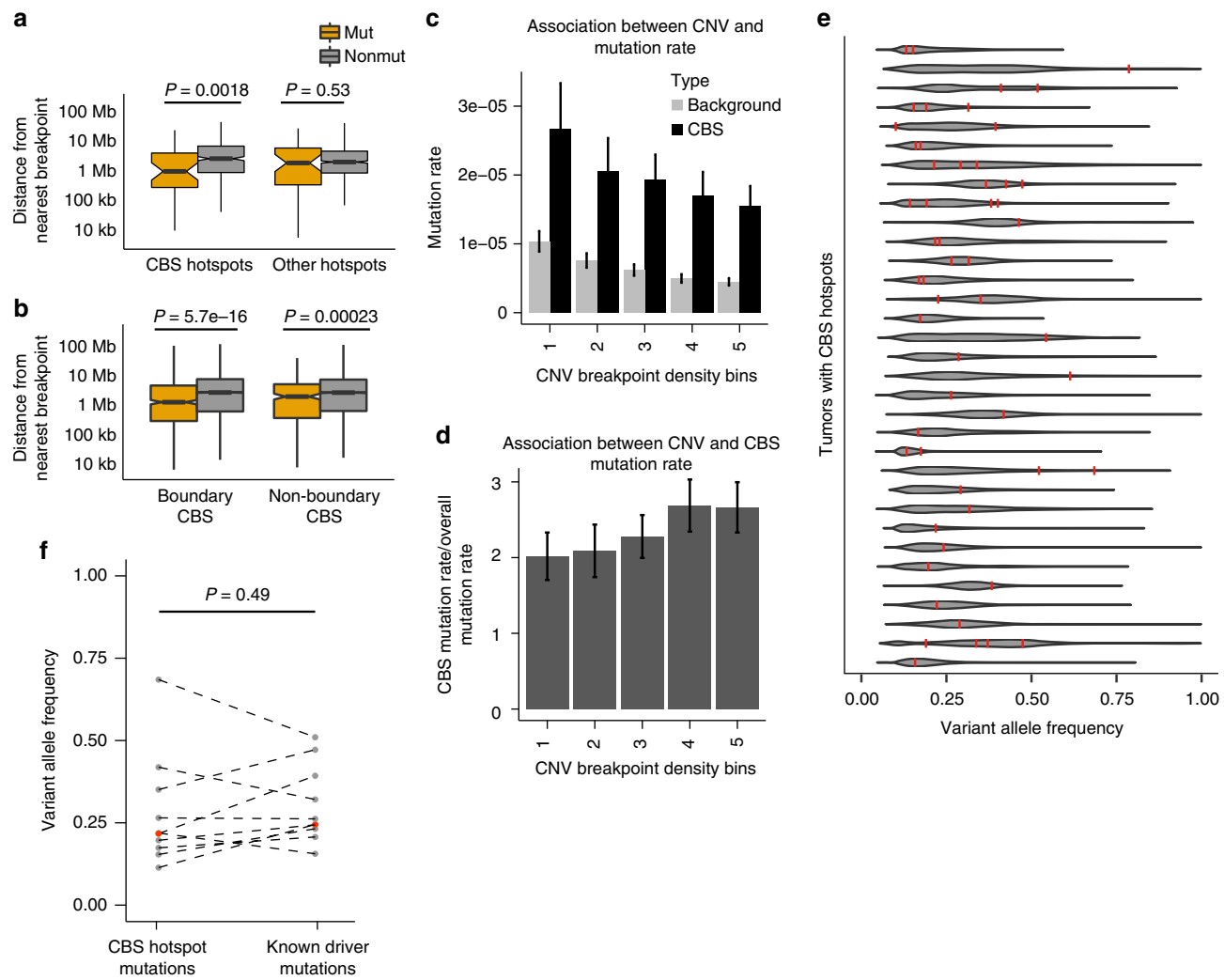

**Fig. 7** Association between CBS mutations and genomic instability. **a** Distance to the nearest CNV breakpoint from CBS hotspots and other non-CBS mutation hotspots. **b** Distance to the nearest CNV breakpoint from CBSs at loop boundary and non-boundary CBSs. Wilcoxon rank-sum test *P*-values are shown. **c** Correlation of mutation rates with SCNA breakpoint density. **d** Correlation of normalized mutation rates with SCNA breakpoint density, correcting for the background mutation rate in each bin. Error bars represent the s.e.m. **e** The violin plots show the VAF distributions of somatic mutations in diploid regions of individual tumors. VAFs of the mutations at CBS hotspots are marked by red vertical lines. **f** Comparison between VAFs of the CBS hotspot mutations and VAFs of non-silent coding mutation on GC driver genes. The red points represent the median VAFs in each group. The dashed lines match mutations from the same samples. *P*-value is calculated by paired Wilcoxon rank-sum test

breakpoints, suggesting a general link between CBS mutations and genomic instability. A previous study has suggested a model where genome higher-order interactions are directly poised for chromosomal breaks[59]. One important open question is whether these processes are coupled, and if so, what is the temporal order of CBS mutations and chromosomal breaks. Interestingly, somatic variant allele frequencies (VAFs) of the CBS hotspot mutations supported that these were generally clonal and likely early events in tumor evolution (Fig. 7e). Furthermore, we found that the VAFs of CBS hotspot mutations were comparable to non-silent coding mutations of known GC driver genes from the same sample (paired Wilcoxon *P*-value = 0.49; Fig. 7f).

Previous studies found kilobase sized regions of hypermutation, termed "kataegis", that tend to co-occur with genomic rearrangements in cancer[44,60]. Importantly, our data suggest that the mutational mechanism underlying the association between CBS mutations and DNA breakpoints is distinct from that of kataegis. While kataegis is characterized by C>T and C>G substitutions, CBS mutations are mostly T>G and T>C substitutions. In addition, kataegis is defined by mutation clusters with inter-

mutation distance <1 kb. CBS hotspots are confined focal regions of <30 bps including the CTCF motif and its 5′ flanking sequence.

Only a subset of tumor samples in our cohort had paired gene expression data (49/187 samples). This limited our ability to test for functional consequences of CBS hotspot mutations. Additional focused experiments involving transcriptome, copy number, and chromatin structure data should further clarify the regulatory and functional effects of the CBS mutations. We did not uncover a shared theme for the 23 significant non-CBS hotspots. Among the non-CBS hotspots, seven of them are intronic, one is downstream of a gene and the rest are intergenic. None of the genes associated with the hotspots are known cancer drivers. We did not observe any mutation hotspot near *TERT*, confirming that the reactivation of *TERT* is very rare in GC[3]. For the non-CBS hotspots that overlapped gene regions, focused functional validation experiments could be performed on a case-by-case basis.

The statistics of cancer driver identification is still limited by our knowledge of the somatic mutation and repair processes. Although our background model corrected for many covariates of

the somatic mutation rate, such as epigenetic and sequence context features, false positives and false negatives could still arise from the current model not considering as yet unknown mutational biases.

Taken collectively, 25% of GC tumors and 19% of colorectal cancer tumors are mutated in at least one of the 11 CBS hotspots. Overall, our analyses nominate these CBS hotspots as potentially common drivers of gastrointestinal cancers. Furthermore, the data supports a general link between CBS mutations and chromosomal instability. This suggests that non-coding regulatory mutations could potentially drive tumor evolution through interfacing with cancer genome and epigenome plasticity.

## Methods

**GC whole-genome sequence data.** We performed whole-genome sequencing of 40 gastric GC tumors and matched normal samples from patients from Singapore (study protocol approved by National University of Singapore Institutional Review Board). Informed consent was obtained from all participating patients. Genomic DNA of tumors and matched normal gastric tissues was extracted (QIAGEN). Libraries were constructed with 300–400 bp insert length, and 101 bp or 151 bp paired-end sequencing was performed on Illumina Hiseq instruments. The tumors were classified into four molecular subtypes as described previously by TCGA[19].

We obtained WGS data of 40 GC tumors from TCGA (https://gdc.cancer.gov), 32 tumors from ICGC (https://ega-archive.org/datasets/EGAD00001003132), and 100 tumors from Wang et al. (HK)[20]. The molecular subtypes of tumors from the TCGA cohort were defined by TCGA. For the HK cohort, only EBV and MSI subtype status was available. The molecular subtypes of tumors from the ICGC cohort were unavailable, but we identified one MSI sample from the ICGC cohort using MSIseq[61].

**Alignment and somatic mutation calling.** Raw sequencing data was uniformly processed using the bcbio-nextgen pipeline (v0.9.3). Briefly, sequencing reads were aligned to the human reference genome (hg19) using BWA[62]. Duplicated reads marked by Picard were removed. Indel regions were realigned using GATK[63]. Somatic mutations were called by four independent mutation callers: VarScan[64], MuTect[65], VarDict[66], and FreeBayes[67] using default parameters of the bcbio-nextgen pipeline. As the nature of our analyses requires high specificity in somatic mutation calling, we developed a random forest predictor, SMuRF[68], trained on manually curated true somatic mutations to identify high confidence somatic mutation calls from the output of the four mutation callers. For each GC WGS sample, a set of high confidence consensus calls were obtained by running the random forest prediction algorithm.

**Additional filters to remove sequencing artefacts.** False positive somatic calls could arise from sequencing and mapping errors. More false positives tend to be called in the non-coding regions of the genome because these regions are enriched for repeats and low sequence complexity regions. As the downstream mutation recurrence analysis is extremely sensitive to recurrent artefacts in somatic mutation calling, we applied additional post-processing filters to eliminate potential false positive calls. We removed candidate somatic mutation calls that:

1. Are found at >1% allele frequency in the 1000 Genomes Project[69] (potential germline mutations)
2. Are found in more than 10% of the matched normal samples (potential systematic sequencing errors)
3. Are found in more than 1% of the matched normal samples and are within 20 bp to a common indel in the 1000 Genomes Project (potential errors arising from mapping errors near indels).

In addition, we removed indel calls that overlap mono-nucleotide repeats of 8 bp or longer. The final set of somatic SNVs and indels are provided in Supplementary Data 3 and 4.

**Gene expression data.** We performed RNA-sequencing on 19 matched tumor-normal pairs. Total RNA was extracted using the Qiagen RNeasy Mini kit. RNA-seq libraries were constructed according to manufacturer's instructions using Illumina Stranded Total RNA Sample Prep Kit v2 (Illumina, San Diego, CA), Ribo-Zero Gold option (Epicenter, Madison, WI), and 1 μg total RNA. We validated the completed libraries with Agilent Bioanalyzer (Agilent Technologies, Palo Alto, CA), and applied the libraries to an Illumina flow cell via the Illumina Cluster Station. RNA-seq reads (2 × 101 bp) were aligned to the human genome (hg19) using TopHat2-2.0.12 (default parameter and—library-type fr-firststrand). Transcript abundances at the gene level were estimated by Cufflinks[70]. The normalized counts of RNA sequencing data of 35 tumors from the TCGA cohort were obtained from the Genomic Data Commons Portal.

**Epigenomic and sequence covariates of somatic mutation rate.** The somatic mutation rate is correlated with epigenetic features such as histone modification and chromatin accessibility[30], especially those derived from the cell type of origin of the cancer[32]. We compiled 36 gastric specific and 24 general chromatin features that potentially affect mutation rate in GC. These 66 histone modification profiles and chromatin accessibility profiles were obtained from Roadmap Epigenomics[29] and in-house data. We obtained P-value signal tracks of 853 DNaseI and histone modification profiles of 111 primary tissues and cell types from the Roadmap Epigenomics project. Among them, 27 epigenetic profiles were derived from gastric related tissues. For the 24 histone marks that were not assayed in gastric-related tissues, we created meta histone modifications profiles by taking the median profile of each mark across all tissues and cell-types assayed. In addition, we included histone modifications profiles of H3K4Me1, H3K4me3, and H3K27Ac of 19 GC tumor/normal samples and 13 GC cell lines (FU97, KATO3, MKN7, NCC24, NCC59, OCUM1, RERF-GC-1B, SNU16, SNU1750, YCC3, YCC7, YCC21, and YCC22)[24,25]. We used the median signal of each histone mark over all tumor samples, all normal samples, and all cell lines, respectively.

Replication timing profiles were not available for gastric tissue. We therefore used the mean replication timing profile of 13 cell lines (Bj, Nhek, K562, Mcf7, Gm06990, Gm12812, Imr90, Hepg2, Helas3, Gm12801, Huvec, Gm12878, and Gm12813) generated by ENCODE[71].

Binding profiles of 132 transcription factors and a meta-profile of all TFBSs were obtained from the Ensembl Regulatory Build[72]. We used generic TF binding profiles as there is no comprehensive TF binding assay done in gastric tissue. In total, we considered 194 candidate epigenetic covariates potentially informative of somatic mutation rates in GC (Supplementary Data 2).

To identify sequence context features affecting somatic mutation accumulation in GC, we considered 1-mer, 3-mer, and 5-mer nucleotide motifs centered at the mutated site, as well as 1-bp and 2-bp left/right flank motifs of the site. All nucleotide context features were grouped into reverse compliment pairs. As indels tend to occur in poly-monomer sequences, especially poly-A and poly-T sequences, we used the presence of poly-A, poly-T, poly-G, and poly-C sequences at the indel sites as features in the indel background mutation model.

Lastly, we included local mutation rate as a covariate to account for other unknown factors affecting mutation rate. The local mutation rate was calculated for 100 kb non-overlapping bins across the genome after masking CDS regions, immunoglobulin loci, and poorly mappable regions (mappability score < 1 in the ENCODE 75mers Alignability track).

**PCA on the epigenetic features.** The genome was divided into 1 Mb non-overlapping windows. CDS regions, immunoglobulin loci, and poorly mappable regions were masked from the genomic windows. Windows smaller than 250 kb after masking were removed. We calculated the mean signal of each epigenetic feature (in Fig. 1b) and the mutation rate of each tumor in each window. The Pearson correlations between the epigenetic features and mutation rates of the each tumor were calculated. To identify the contributions of epigenetic features to the variance in the mutation rate of individual tumors, we performed PCA on the correlation matrix between the mutation rates of individual tumors and epigenetic features using the "prcomp function in R. The contribution of each feature to a principal component is calculated as the feature's loading (rotation) divided by the sum of loadings of all features for that principal component.

**Feature selection using LASSO regression.** The LASSO is a regularized regression approach commonly used for automated feature selection. LASSO penalizes the sum of the absolute size of the regression coefficients, forcing some of the regression coefficients to shrink to zero, thereby selecting a simpler and more interpretable model. The LASSO objective function can be written as:

$$\min_{\beta_0, \beta} \frac{1}{N} \sum_{i=1}^{N} l(y_i, \beta_0 + \beta^T x_i) + \lambda \parallel \beta \parallel_1$$

Where $l$ is the negative log-likelihood function and $\lambda$ is the regularization parameter.

We used LASSO logistic regression to identify the most informative features for modeling the somatic mutation rate in GC. As it is computationally expensive to run a logistic regression on all positions in the non-coding genome with a large number of predictor variables, we used all mutated sites and an equal number of randomly sampled non-mutated sites as the input for feature selection in the LASSO logistic regression model. We regressed the binary mutation status of each site against the mean signal of each feature over an 11 bp region centered at the site. The regularization parameter $\lambda$ was chosen by 10-fold cross-validation such that the error of the selected model was within one standard deviation from the minimum error. LASSO regression and cross validation were performed using the "glmnet" package in R.

$$\text{glmnet}(y \sim \beta X, \text{family} = \text{logistic})$$

We bootstrapped 100 samples with 50% of the data at each bootstrap, and performed LASSO regression using the bootstrap samples. Assuming that the most

informative features would be robustly selected, we used features selected in more than 95% of the bootstrap samples for the final regression model.

**Tumor-specific background mutation model**. The patient specific background mutation probabilities were estimated by fitting a logistic regression model on all genomic sites after masking CDS regions, immunoglobin loci, and poorly mappable regions. Replication timing was discretized into eight equally sized bins, the local mutation rate was discretized into ten equally sized bins, and the chromatin features and TF-binding profiles were binarized. $P$-value signal tracks of the histone modification profiles from the Roadmap Epigenomics were binarized using a cutoff of $10^{-4}$. ENCODE TF-binding profiles were binarized according to the presence of a peak in any cell line assayed. We performed logistic regression using the frequency table of the counts of mutated and non-mutated sites for each combination of the covariates. Separate logistic regression models were fit to estimate the background mutation probabilities of SNVs and indels. This is to account for the different mutational processes from which SNVs and indels arise, as well as the different uncertainties associated with SNV and indel calls.

$$\mathrm{glm}(y \sim \mathrm{rep} + \mathrm{epi} + \mathrm{sequence} + \mathrm{pid}, \mathrm{family} = \mathrm{logit})$$

Here rep is the Repli-seq profile, epi represents the epigenomic features, sequence represents the sequence context features, and pid is the patient ID. Features used in each model are shown in Supplementary Fig. 2.

**Poisson binomial model of mutation recurrence**. For a specific region of interest, the probability, $p_i$, of mutation in tumor $i$ is a function of the length of that region and the expected mutation rates of individual nucleotides in that region under the null hypothesis. Assuming $q_{i,j}$ is the mutation probability of nucleotide $j$ in tumor $i$, and $l$ is the length of the region of interest:

$$p_i = 1 - \prod_{j=0}^{l}(1 - q_{i,j})$$

Mutation recurrence is then modeled using the Poisson binomial distribution, which accounts for variation in mutation rate across tumors. For a specific region of interest, the probability of having mutations in $k$ or more individuals is given by:

$$\Pr(K \geq k) = \sum_{m=k}^{n} \sum_{A \in F_m} \prod_{i \in A} p_i \prod_{j \in A^c}(1 - p_j)$$

Here $n$ is the total number of tumors sequenced, $k$ is the number of tumors with mutations in the region of interest, $F_m$ is the set of all subsets of $k$ integers selected from $\{1,2,…,n\}$, $A$ is a subset of $F_m$, $A^c$ is the complement of set $A$, $p_i$ is the probability of mutation in tumor $i$, and $p_j$ is the probability of mutation in tumor $j$. The Poisson binomial probability is calculated using an efficient and accurate normal approximation in the "poibin" R package.

**Identification of mutation hotspots**. The hotspot analysis aims to identify small focal regions with high mutation rates. We first considered all mutated 21 bp regions by taking 10 bp flanks on each side of each mutation. Then we calculated the mutation recurrence scores for all 21 bp regions with three or more mutated samples (two or more for indels). The $P$ value of mutation recurrence of each hotspot was calculated using the Poisson binomial model described in the previous section. The total number of hypothesis tested is equal to the number of bases in the masked non-coding genome. We used the Bonferroni correction to adjust for multiple testing of 2,533,374,732 hypotheses, to maintain the overall $\alpha$ at 0.01.

**Identification of non-coding regions with indel recurrence**. We scanned for non-coding regions of genes with recurrence of indels. Gene regions were defined by Ensembl v75 annotations. We considered the merged non-coding regions of each gene by masking all coding regions of each gene, and extending the gene boundaries by 1 kb to take into account its promoter region. We calculated the mutation recurrence scores for all protein-coding genes, and their individual merged non-coding regions, using the Poisson binomial model described in the previous section. The Bonferroni correction was used to maintain the overall $\alpha$ at 0.01.

**Enrichment of mutation hotspots in functional regions**. We calculated the log odds ratio of the enrichment of hotspot mutations in TF binding regions and conserved DNA elements. Gastric-specific TFBSs were defined as a ChIP-seq peak of a TF in any of the ENCODE cell lines that overlaps a gastric tissue DNaseI hypersensitivity site (data from Roadmap Epigenomics). Constitutive TFBSs are defined as TFBSs with $P_{\mathrm{tfbs}} > 0.75$, where $P_{\mathrm{tfbs}}$ is the probability that the TFBS is bound by a TF for any given ENCODE cell line. $P_{\mathrm{tfbs}}$ for all TFBSs were obtained from the ENSEMBL regulatory build. Conserved elements generated by GERP[73]

from the alignment of hg19 to 36 mammals were downloaded from the UCSC genome browser.

The expected fraction of hotspot (or non-hotspot) mutations in the functional region type ($p_2$) is the fraction of the genome that constitutes the functional region. The observed fraction of hotspot (or non-hotspot) mutations in the functional region is calculated by adding all mutations in the functional region type and dividing by the total number of mutations genome-wide ($p_1$). The log odds ratio of the enrichment of hotspot (or non-hotspot) mutations in a functional region type is given by,

$$\mathrm{LOD} = \ln\left(\frac{p_1/(1-p_1)}{p_2/(1-p_2)}\right)$$

The standard error of the LOD is calculated as,

$$\mathrm{SE}_{\mathrm{LOD}} = \sqrt{\frac{\mathrm{SE}_{p_1}^2}{p_1^2 - (1-p_1)^2} + \frac{\mathrm{SE}_{p_2}^2}{p_2^2 - (1-p_2)^2}}$$

The statistical significance of the enrichment was evaluated by the $Z$-test.

**Identification of gastric-specific CBSs**. The position weight matrix of the CTCF binding motif was obtained from JASPAR[74]. Genomic locations of CTCF binding motifs were identified using the FIMO[75] function of the MEME tool suite[76] with a $P$-value threshold of 0.01. Gastric-specific CBSs were defined as CBS motifs overlapping both a CTCF ChIP-seq peak in at least one ENCODE cell line and a DNaseI hypersensitivity site in gastric tissue from Roadmap epigenomics. We used the set of constitutive CTCF–CTCF loops shared across three cell lines (GM12878, Jurkat, and K562) obtained from the supplementary information of Hnisz et al.[13] CBSs that overlap the boundaries of these constitutive CTCF loops were defined as boundary CBSs.

**The CBS-specific background model**. For the CBS-specific background model, we limited the model and search space to CBS regions and their 5 bp flanking DNA.

$$\mathrm{glm}\big(y_{\mathrm{CBS}} \sim \mathrm{rep} + \mathrm{subtype} + \mathrm{boundary} + \mathrm{sequence} + \mathrm{pid}$$
$$+ \mathrm{mutsig1} + \mathrm{mutsig17}, \mathrm{family} = \mathrm{logit}\big)$$

Here subtype is the tumor subtype, boundary indicates if the CBS is located at a CTCF loop boundary, and mutsig1 and mutsig17 represent the percentage contributions of signature 1 and signature 17 of the tumor. We used deconstructSigs[77] to quantify the prevalence of each of the 30 COSMIC consensus mutation signatures in each tumor.

The $P$ value of mutation recurrence of each CBS was calculated using the Poisson binomial model described in the previous section. The Bonferroni correction was applied to maintain the overall $\alpha$ at 0.01.

**Motif analysis of hotspot mutations in the CTCF motif flanks**. We extracted the ±40 bp sequence context around each mutation, and used DeepBind to predict the binding scores of 472 TFs for the reference (ref score) and mutated sequences (alt score) of each mutation. Since the binding scores output by DeepBind are on an arbitrary scale and vary between different TF models, we estimated the background distributions of the binding scores of each TF by applying DeepBind to 10,000 randomly sampled non-hotspot mutations. For a particular TF, a mutation is predicted to be motif-disrupting if its reference sequence scores higher than 99.9% of the random mutations, and the score difference between its alternate and reference sequences (alt score – ref score) is smaller than 99.9% of the random mutations for that TF. A mutation is predicted to create a motif for a specific TF if its alternate sequence scores higher than 99.9% of the random mutations, and the score difference between its alternate and reference sequences (alt score – ref score) is greater than 99.9% of the random mutations for that TF.

**Pan-cancer analysis of mutation recurrence at CBS hotspots**. Somatic mutations of 858 tumors from 22 cancer types were downloaded from the supplementary information of Weinhold et al.[5] Hypermutated tumors with more than 200,000 mutations were excluded from the analysis. Cancer types with less than ten samples were excluded from the analysis. For CBS mutation rate calculation in Fig. 6b, CBSs were defined as CTCF motifs overlapping a CTCF ChIP-seq peak in at least one ENCODE cell line. We further defined tissue-specific CBSs for 14/19 cancer types for which DNaseI profiles in the matched tissue types are available in Roadmap Epigenomics. Tissue-specific CBSs were defined as generic CBSs that fall under DNaseI peaks in the respective tissue. Supplementary Figure 15 shows the mutation rates at tissue-specific CBSs.

**Analysis of SCNA breakpoints**. Copy number segmentations were generated by CNVkit[78] using default settings (bcbio-nextgen v0.9.3). SCNA breakpoints were defined as the ends of non-diploid segments. Assuming tumor purity of 50%, the estimated mean purity of these tumors, non-diploid segments were defined as

segments with log2(tumor coverage/normal coverage) < log2(1.5/2) or log2(tumor coverage/normal coverage) > log2(2.5/2).

**Analysis of VAFs**. The list of known GC driver genes was collated from the Cancer Gene Census[79] and the driver genes identified by TCGA[19] and Wang et al.[20] We excluded *TP53* from the analysis as *TP53* frequently undergo deletions and loss of heterozyosity. We identified non-synonymous and truncating mutations on known GC driver genes, and compared their VAFs to the VAFs of CBS hotspot mutations from the same samples using a matched Wilcoxon rank-sum test. Only mutations in diploid regions in each sample were included in the analysis.

**Code availability**. All R code used to generate the figures and statistics of the paper is included in Supplementary Data 5. Source code for the ensemble somatic mutation caller, SMuRF[68], can be found at https://github.com/skandlab/SMuRF. Source code for estimating background mutation rate from genomic covariates and identification of non-coding mutation hotspots is available at: https://github.com/skandlab/MutSpot.

**Data availability**. SG tumor data: Sequence data has been deposited at the European Genome–phenome Archive (EGA), which is hosted by the EBI and the CRG, under accession number EGAS00001002872.
TCGA tumor data: https://portal.gdc.cancer.gov/projects/TCGA-STAD
ICGC tumor data: https://ega-archive.org/datasets/EGAD00001003132
HK tumor data: https://ega-archive.org/datasets/EGAD00001000782
Roadmap Epigenomics data: http://www.roadmapepigenomics.org/data/
Encode data: ftp://ftp.ensembl.org/pub/release-85/regulation/homo_sapiens/.

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

## Acknowledgements

This study makes use of data generated by the Department of Pathology of the University of Hong Kong and Pfizer Inc. A full list of the investigators who contributed to the generation of the data is available from *Nature Genetics* 2014: 46(6):573-82, doi:10.1038/ng.2983, Epub 2014 May 11. This work was supported by Singapore National Medical Research Council grants OFIRG15nov072, TCR/009-NUHS/2013, and NMRC/STaR/0026/2015.

## Author contributions

A.J.S., P.T., and Y.A.G. designed the study. Y.A.G. and M.M.C. analyzed the data. W.H. performed mutation calling. M.X. and W.F.O. contributed data analysis. Y.A.G., A.J.S., and P.T. interpreted the data and wrote the manuscript, with contributions from all authors.

## Additional information

**Competing interests:** The authors declare no competing interests.

