## [Peer Review File · Nature Communications]

Reviewers' comments:

Reviewer #1 (Remarks to the Author):

Guo et al. have made several changes to their manuscript, which I now see to be much improved compared to their previous submission to Nature Genetics, and the study presents overall an interesting insight into the genomics of gastric cancer, with novel results being presented in a cutting-edge area of cancer genetics.

There are still a few points that I do not think have been sufficiently addressed:

1. Regarding my first question (reviewer 1) about finding elevated mutation rates in 21bp bins: The authors acknowledge that their window size is rather small and that it does not capture both motif sites as well as flanking regions at the same time. They claim to have done the same analysis using 25bp bins (but not 40bp bins as I suggested) and found "7 additional significantly mutated CBSs". What about the hotspots that they found in the 21bp analysis - which they present unchanged in the manuscript? Are these hotspots still present when increasing the window size to 25bp? Given that there is no evidence presented that the identified hotspots remain when the parameters are relaxed to include more sites, I am still sceptical of this main result.
2. Related to the above, there could be more of a discussion as to how putative hotspot mutations in the flanking regions of CTCF binding sites would drive changes in gene expression.
3. Regarding my previous comment number 6: the authors find a depletion of non-hotspot mutations at TFBSs, which is in contrast to previous studies that found an overall enrichment at such sites (albeit in different cancer types). The authors could highlight more the fact that they didn't use tissue-matched predictions of TFBSs, and I would assume that a lot of the TFBSs that they use are not actually bound in gastric cancer. Ideally, they could calculate the enrichment for constitutively bound TFBSs - even if they do not have data for GC, constitutive TFBSs might be more conservative. Related to that: I assume that mapping bias/mappability of mutations was taken into account in their analysis although I do not see this mentioned in the text.

Reviewer #2 (Remarks to the Author):

The responses to my comments, although carefully considered, do not fully address two points.

1. That the mutations in CBSs are under positive selection. I agree that these mutations were "selected" in the sense of a passenger mutation, but I think that in cancer, positive selection implies driver mutations. This has not been shown in this study. I see two alternatives, either experiments to show that, which will result in more time and cost, or tone this down in the text.
2. Expression analysis. I stand behind my initial comments. I like the example that the authors mentioned. In deed the levels of TERT expression has not been that different between mut and wt (this has explanations that are not necessary for this review). However, TERT mutations are in 2 positions (the vast majority of the time). The two mutations are mutually exclusive, they correlate well with the disease (sporadic or familial), now it is known which TFs binds to them and that the mutant allele is the one that is expressed.

There are some issues that need clarification. The text writes, "The first hotspot we identified is located in a CBS on chromosome 6 and has mutations in 12 samples (Fig. 5a-c). The expression of two neighboring genes, CENPQ and MUT, ~1Mb upstream of this hotspot was significantly elevated in the mutated samples ($P=0.007$ and 0.0021 respectively, adjusted $P=0.026$ and 0.042 respectively, two-sided Wilcoxon rank-sum test; Fig. 5a-c)". Figure 5c shows 3 out of the 12 samples with mut. Two of the three are within the values for the wt. Can you show on this figure the 3 mutations in the CTCF motif that are relevant to the CENPQ expression? Same for the other panels. Is there a SNP in any of these genes that can help you identify if the two alleles and try to correlate a heterozygote DNA mutation in the CBS with expression of one of the alleles? Or, some other experiment in a another sample to show that there is some cause an effect.

Overall all I think that these two issues need to either toned down (this could make the manuscript less novel), or provide more evidence.

Reviewer #3 (Remarks to the Author):

The manuscript now reads better as the authors have clarified some issues. They used their statistical pipeline to find CBS hotspots but most of the mutations are outside the CTCF motif so it is less intuitive to think that they are functional. One way to address this is to see if there are motifs for other TFs at positions -5 to 0 in the CBS hotspots or all CBS mutations in Fig 4f. The response, only repeats arguments from the previous version of the manuscript.

This reviewer asked for more evidence that the CBS hotspots are functional, than what was presented regarding changed expression of nearby genes. This analysis now has been revised and it turns out that after correction for multiple testing the change in expression for SPG20 is no longer significant. This should be made clear to the reader so in the section "CBS hotspot mutations alter expression of neighboring genes" they should write "We found genes with nominally altered expression for 3 of the 4 hotspots" and in Discussion "Out of the 4 CBS hotspots examined, 3 of them were associated with nominally significant expression changes.....". In Fig 5 c, f and I they still show the nominal p-values which is not appropriate. They should either change to the values corrected for multiple testing or show both. They still have an argument that the genes change in the same direction in wt tumors as compared to normal tissue so the finding is suggestive but should be presented in a more objective way.

This reviewer also asked if the CBS mutations have been seen before in other studies, regardless of any evidence of positive selection but apparently the authors misunderstood the question. It would be interesting to know if the mutations have been seen before in other tumors and this can be presented in a supplementary table and commented on in the text.

It is still an issue that they find association between CBS mutations and SCNA only in CIN tumors even though they are present also in GS tumors (Fig 4b). I wonder if they could clarify this by repeating the analysis in CIN presented in Fig 7 c, d also in GS tumors.

Finding non-coding mutations that contribute to cancer is a challenging but interesting area, which the scientific community is getting better at but the methods can still improve, whether statistical or based on wet lab experiments. It is therefore appropriate to comment on this, perhaps at the end of the paper, with a discussion on the fact that we do not know all biases that exist in the processes of mutation and repair, which gives limitations to the statistical methods. This is also warranted by the fact that the 23 non-CBS hotspots never co-localized with TF-binding regions which makes the biological interpretation less logical.

Point-by-point Response Letter

Reviewer 1, Comment 1: “the study presents overall an interesting insight into the genomics of gastric cancer, with novel results being presented in a cutting-edge area of cancer genetics ...” 1

Reviewer 1, Comment 2: Are 21bp windows adequate to identify CBS hotspots? .. 2

Reviewer 1, Comment 3: How hotspot mutations in the flanking regions of the CTCF motif drive gene expression changes? 3

Reviewer 1, Comment 4: Using tissue-specific or constitutive TFBSs for enrichment analysis..... 6

Reviewer 2, Comment 1: “carefully considered” response..... 7

Reviewer 2, Comment 2: Justify the use of positive selection 8

Reviewer 2, Comment 3: Conclusions from the mRNA expression analysis 9

Reviewer 3, Comment 1: “The manuscript now reads better as the authors have clarified some issues.” 12

Reviewer 3, Comment 2: Functional roles of hotspot mutations in the flanking regions of the CTCF motif 12

Reviewer 3, Comment 3: Present expression analysis more objectively 15

Reviewer 3, Comment 4: Previous validations on CBS mutations 15

Reviewer 3, Comment 5: Association between SCNA and CBS mutations in the GS subtype..... 16

Reviewer 3, Comment 6: Add discussion on the limitations of the statistical methods in non-coding driver discovery 17

Reviewer 1, Comment 1: “the study presents overall an interesting insight into the genomics of gastric cancer, with novel results being presented in a cutting-edge area of cancer genetics ...”

Reviewer Comment	Guo et al. have made several changes to their manuscript, which I now see to be much improved compared to their previous submission to Nature Genetics, and the study presents overall an interesting insight into the genomics of gastric cancer, with novel results being presented in a cutting-edge area of cancer genetics. There are still a few points that I do not think have been sufficiently addressed:
Author Response	We would like to thank all reviewers for their appreciation of our work and for providing constructive suggestions to improve the manuscript.

Reviewer 1, Comment 2: Are 21bp windows adequate to identify CBS hotspots?

Reviewer Comment	Regarding my first question (reviewer 1) about finding elevated mutation rates in 21bp bins: The authors acknowledge that their window size is rather small and that it does not capture both motif sites as well as flanking regions at the same time. They claim to have done the same analysis using 25bp bins (but not 40bp bins as I suggested) and found “7 additional significantly mutated CBSs”. What about the hotspots that they found in the 21bp analysis - which they present unchanged in the manuscript? Are these hotspots still present when increasing the window size to 25bp? Given that there is no evidence presented that the identified hotspots remain when the parameters are relaxed to include more sites, I am still skeptical of this main result.
Author Response	The reviewer is concerned that 21bp windows are not sufficient to cover both the CTCF motif and the flanks. As previously mentioned, in our CBS specific model, we tested 29bp regions, which included the 19bp CBS motif, 5bp 3' flank and 5bp 5' flank of the motif (see manuscript pages 15 and 35). In this analysis, 9/11 CBS hotspots identified from the genome-wide analysis remain significant ($P < 0.01$, Bonferroni correction), and the other 2 are marginally significant with adjusted P-values of 0.025 and 0.086 (page 15). To further address the reviewer's question, we performed a genome-wide hotspot analysis with 41bp windows as suggested. As expected, the top-ranked hotspots are generally concordant between the 21bp and 41bp-window analysis. 17/34 hotspots remain significant at the Bonferroni corrected significance threshold of 0.01, and all are significant with a 1% FDR (Supplementary Fig. 4). We identify 2 and 16 new hotspots at Bonferroni $P=0.01$ and 1% FDR, respectively. Overall, this suggests that 21-bp windows are adequate to capture most focal mutation clusters in gastric cancer. Furthermore, as most mutation hotspots cluster within 21bp, increasing the window size leads to lower significance levels for most hotspots and decreases the sensitivity of hotspot identification.
Changes to manuscript	[Page 11] To test if the 21bp window size was adequate to capture most mutation hotspots, we repeated the hotspot analysis using larger 41bp windows. In general, the rankings of the hotspots remained stable (Supplementary Fig. 4). 17/34 hotspots remained significant and only 2 additional hotspots were identified ($P < 0.01$, Bonferroni correction). [Supplementary Fig. 4]

Reviewer 1, Comment 3: How hotspot mutations in the flanking regions of the CTCF motif drive gene expression changes?

Reviewer Comment	Related to the above, there could be more of a discussion as to how putative hotspot mutations in the flanking regions of CTCF binding sites would drive changes in gene expression.
Author Response	The reviewer raises an interesting question on the functional consequences of the hotspot mutations in the flanks of the CTCF binding motifs. A previous study found that the flanking sequences of weaker CTCF binding sites (those with lower match scores to the CTCF positional weight matrix) are more conserved and could be important for context-specific CTCF binding at these sites (Essien et al., Genome Biol, 2009, doi: 10.1186/gb-2009-10-11-r131). The same observation was also found to be true for REST binding sites (Bruce et al., Genome Res, 2009, doi:10.1101/gr.089086.108). It is possible that the hotspot mutations in the flanking sequences of the CTCF motifs could affect CTCF binding by altering the sequence context of the CBSs. We examined the evolutionary constraints (PhyloP scores based on placental mammals) in the flanking sequences of the mutated CBSs. In general, the 5' flanks of the CBS motif were not conserved, but the 3' flanks of the CBS motif tended to be conserved (Supplementary Fig 5a). Interestingly, at the hotspot upstream of CENPQ, there is a cluster of mutations at the 5' flank of the CTCF motif and the mutations coincide with a cluster of conserved bases. Therefore, the 5' flank of this hotspot is under evolutionary constraint and is likely important for

	CTCF function at this site. In addition, we found another CBS hotspot where 9 mutations in the 5' flank of the CBS motif coincided with a highly conserved based (Supplementary Fig. 13). Another potential explanation is that the hotspot mutations in the flanks of the CTCF motifs create/disrupt binding sites for other transcription factors. To examine this possibility, we used DeepBind to predict the effect of these mutations on transcription factor binding. We only found a few CBS sites with predicted change in transcription factor binding. Mutations in flanks of 2 hotspots were predicted to create binding sites for ATF2 and RCOR1. Mutations in the flank of another CBS hotspot were predicted to disrupt the binding of SIN3A (Supplementary Table 6). Finally, it is also possible that some of the mutations in the flanking regions of the CTCF motifs are just passenger mutations that arise due to the elevated mutation rates at CBSs. While our model allows identification of individual CBS with overall mutation enrichment, it does not allow us to distinguish between passenger and driver mutations within such a region. In the revised manuscript, we have added the evolutionary conservation analysis in Supplementary Fig. 13, motif creation/disruption analysis in Supplementary Table 6, and added a paragraph in the discussion on the potential roles of the CBS hotspot flank mutations.
Changes to manuscript	[Page 18] Many of the hotspot mutations were located in the 5' flanks of the consensus CTCF motif (Fig. 4f). Previous studies have found increased conservation of the flanking sequences of weaker CTCF and REST binding sites, suggesting that the sequence context is important for TF binding at these sites^{51, 52}. We examined evolutionary conservation⁵³ at the CTCF binding motifs and their flanking sequences. In general, the 5' flanks of the CTCF motifs were not conserved (Supplementary Fig. 13a). However, in the hotspot upstream of CENPQ, the mutation cluster in the 5' flank co-occurred with conserved bases (Supplementary Fig. 13b). In addition, we found another CBS hotspot with 9 5'-flank mutations that coincided with a highly conserved base (Supplementary Fig. 13c). Such hotspot mutations, affecting conserved 5' flanks of CTCF motifs, could disrupt context-specific binding of CTCF. We also examined the possibility that mutations in the flanking regions of CTCF motifs create or disrupt binding motifs of other TFs. We used DeepBind⁵⁴ to predict the binding scores of wildtype and mutated sequences for 472 transcription factors. However, we only found mutations at three CBS sites with predicted change in TF binding

(**Supplementary Table 6**). Lastly, it is also possible that some mutations at CBS flanks are passenger mutations arising due to the overall elevated mutation rates at CBSs. While our model identifies individual CBS regions with overall mutation enrichment, it does not allow us to distinguish between passenger and driver mutations within such regions.

[Supplementary Fig. 13]

Evolutionary conservation of the consensus CTCF motif and flanking sequences. (a) Average PhyloP scores of the CTCF-binding motif and ± 5 flanking bases of all mutated CBSs. (b-c) Two CBS hotspots (b is hotspot upstream of *CENPQ*) where mutations at 5' flanks of CTCF-binding motifs coincide with conserved bases.

[Supplementary Table 6]

DeepBind analysis of hotspot mutations in the flanking regions of

	CTCF-binding motifs
--	---------------------

Reviewer 1, Comment 4: Using tissue-specific or constitutive TFBSs for enrichment analysis

Reviewer Comment	Regarding my previous comment number 6: the authors find a depletion of non-hotspot mutations at TFBSs, which is in contrast to previous studies that found an overall enrichment at such sites (albeit in different cancer types). The authors could highlight more the fact that they didn't use tissue-matched predictions of TFBSs, and I would assume that a lot of the TFBSs that they use are not actually bound in gastric cancer. Ideally, they could calculate the enrichment for constitutively bound TFBSs – even if they do not have data for GC, constitutive TFBSs might be more conservative. Related to that: I assume that mapping bias/mappability of mutations was taken into account in their analysis although I do not see this mentioned in the text.
Author Response	We did in fact use tissue-matched predictions of TFBSs, but we apologize for missing out this detail in the methods and have edited the methods section to make this clear to the reader. In the TFBS enrichment analysis, we used tissue-matched TFBS by overlapping ENCODE TFBSs with DNaseI profiles from gastric tissue. In addition, the reviewer made a good suggestion to check if the enrichment patterns remain the same for constitutive TFBSs. Accordingly, we identified constitutive TFBSs as TFBSs with $P_{tfbs} > 0.75$, where P_{tfbs} is the probability that the TFBS is bound by a TF for any given ENCODE cell line. P_{tfbs} for all TFBSs were obtained from the ENSEMBL regulatory build. The results of our enrichment analysis remains the same using this set of constitutive TFBSs (Supplementary Fig. 3) We agree with the reviewer that it is important to take mappability into account when analyzing somatic mutations. We did this by masking poorly mappable regions (mappability score < 1 in the ENCODE 75mers Alignability track) from the genome, thereby excluding all mutations in poorly mappable regions from our analysis. In the revision, we modified the manuscript to make it clearer that all mutations in poorly mappable regions were removed from the analysis (page 5).
Changes to manuscript	[Page 5] Somatic mutations in CDS regions, immunoglobulin loci and poorly mappable regions were also removed from further analyses.

[Page 11]

Furthermore, we observed a depletion of somatic mutations at gastric-specific TFBSs among the non-hotspot mutations (**Fig. 3b**). Overall, gastric tissue TFBSs comprises ~1% of the genome, but only 0.58% of the non-hotspot mutations were located in these regions. A similar depletion of mutations was observed for constitutive TFBSs (**Supplementary Fig. 3**).

[Page 35]

We calculated the log odds ratio of the enrichment of hotspot mutations in TF binding regions and conserved DNA elements. Gastric-specific TFBSs were defined as a ChIP-seq peak of a TF in any of the ENCODE cell lines that overlaps a gastric tissue DNaseI hypersensitivity site (data from Roadmap Epigenomics). Constitutive TFBSs are defined as TFBSs with $P_{tfbs} > 0.75$, where P_{tfbs} is the probability that the TFBS is bound by a TF for any given ENCODE cell line. P_{tfbs} for all TFBSs were obtained from the ENSEMBL regulatory build.

[Supplementary Fig. 3]

Log odds ratio of the enrichment of hotspot mutations and non-hotspot mutations in constitutive transcription factor binding regions. Error bars indicate the s.e.m of the log odds ratio.

Reviewer 2, Comment 1: “carefully considered” response

Reviewer Comment	The responses to my comments, although carefully considered, do not fully address two points.
------------------	--

Author Response	We would like to thank all reviewers for their appreciation of our work, and for providing constructive suggestions to improve the manuscript.
-----------------	--

Reviewer 2, Comment 2: Justify the use of positive selection

Reviewer Comment	That the mutations in CBSs are under positive selection. I agree that these mutations were “selected” in the sense of a passenger mutation, but I think that in cancer, positive selection implies driver mutations. This has not been shown in this study. I see two alternatives, either experiments to show that, which will result in more time and cost, or tone this down in the text.
Author Response	We agree with the reviewer that although we have statistical evidence for positive selection at these hotspots in gastric cancer, additional experimental evidence is needed to confirm that these mutations are indeed drivers. As mentioned in our previous response, the presence of recurrent mutations beyond chance expectations is generally deemed as statistical evidence of positive selection in the field (Lawrence et al., Nature, 2013, doi:10.1038/nature12213; Watson et al, Nat Rev Genet, 2013, doi:10.1038/nrg3539; Khurana et al., Nat Rev Genet, 2016, doi:10.1038/nrg.2015.17). However, it was never our intention to classify the hotspots as more than candidate driver mutations. To avoid any confusion, we have removed the term “positive selection” from the title of the manuscript, new title: “Mutation hotspots at CTCF binding sites coupled to chromosomal instability in gastrointestinal cancers”. We have also carefully edited the manuscript to clarify that our analyses nominate these hotspots as candidate drivers in gastric cancer, and more work is needed to establish the functional roles of these mutations in gastric cancer tumorigenesis (see below for precise changes to manuscript).
Changes to manuscript	[New title] Mutation hotspots at CTCF binding sites coupled to chromosomal instability in gastrointestinal cancers [Page 4] Overall, our analyses nominate these CBS hotspots as candidate drivers of GC. [Page 15]

	Mutations at these specific sites can therefore not be explained by a genome-wide elevated mutation rate at CBS, indicating that mutations at these focal sites may be positively selected in gastric tumors. [Page 24] Overall, our analyses nominate these CBS hotspots as potential drivers in GC, and support the hypothesis that driver mutations may arise as a by-product of the increased mutation load at CBSs followed by positive selection at specific CBSs. [Page 26] The statistics of cancer driver identification is still limited by our knowledge of the somatic mutation and repair processes. Although our background model corrected for many covariates of the somatic mutation rate, such as epigenetic and sequence context features, false positives and false negatives could still arise from the current model not considering such unknown mutational biases. Taken collectively, 25% of gastric cancer tumors and 19% of colorectal cancer tumors are mutated in at least one of the 11 CBS hotspots. Overall, our analyses nominate these CBS hotspots as potentially common drivers of gastrointestinal cancers. Furthermore, the data supports a general link between CBS mutations and chromosomal instability. This suggests that non-coding regulatory mutations could potentially drive tumor evolution through interfacing with cancer genome and epigenome plasticity.
--	---

Reviewer 2, Comment 3: Conclusions from the mRNA expression analysis

Reviewer Comment	Expression analysis. I stand behind my initial comments. I like the example that the authors mentioned. In deed the levels of TERT expression has not been that different between mut and wt (this has explanations that are not necessary for this review). However, TERT mutations are in 2 positions (the vast majority of the time). The two mutations are mutually exclusive, they correlate well with the disease (sporadic or familial), now it is known which TFs binds to them and that the mutant allele is the one that is expressed. There are some issues that need clarification. The text writes, “The first hotspot we identified is located in a CBS on chromosome 6 and has mutations in 12 samples (Fig. 5a-c). The expression of two neighboring genes, CENPQ and MUT, ~1Mb upstream of this hotspot was significantly elevated in the mutated samples (P=0.007 and 0.0021 respectively, adjusted P=0.026 and 0.042 respectively, two-sided Wilcoxon rank-sum test; Fig. 5a-c)”. Figure 5c shows 3 out of the 12 samples with mut. Two of the three are within the values for the wt. Can you show on this figure the 3 mutations in the CTCF motif that are relevant to the CENPQ expression? Same for the other panels. Is there a SNP in any of these genes that can help you identify if the two alleles and try to correlate a heterozygote DNA mutation in the CBS with expression of one of
--

	the alleles? Or, some other experiment in another sample to show that there is some cause an effect.
Author Response	We agree with the reviewer that more evidence is needed to confirm that these are indeed drivers, like in the case of TERT promoter mutations. The reviewer made a good suggestion to show the mutations that were available for use in the expression analysis. Accordingly, we have updated Fig. 5 to highlight mutations that were used in the expression analysis. For the CENPQ hotspot, the sample with the highest CENPQ expression was mutated at the conserved position 9 of the CTCF motif, while the other two samples were mutated at position 2 of the CTCF motif (one of which also has an additional mutation in the 5' flank of the CTCF motif). It is likely that different mutations in the same hotspot have different disruptive potentials. However, with just 3 mutated samples, it is not possible for us to conclude if the variability in CENPQ expression change is due to mutations occurring at different positions within the CBS, or due to differences in the genetic, epigenetic, or transcriptomic background of these samples. The reviewer made an interesting suggestion to check for allele-specific expression changes at the 3 candidate hotspots. The CBS hotspots are 0.2-1mb away from the candidate genes, and it is much more challenging to perform allele-specific expression analysis at such distant sites compared to for example promoter mutations. We examined the SNP profiles between each of the 3 candidate CBS hotspots and their associated genes in all mutated samples. We found that the maximum inter-SNP distance is >5kb in all mutated samples. Since the samples were sequenced by paired-end sequencing with read lengths of ~100bp and inner-mate distance of ~500bp, it is not possible for us to unambiguously resolve the two alleles and determine the origin of the RNA-seq reads.
Changes to manuscript	[Fig. 5]

Association of CBS hotspot mutations and cis-gene expression. **(a,d,g)** Association between mutation status of the CBS hotspot and expression levels of neighboring genes (two-sided Wilcoxon rank-sum test). Upregulated genes are shown above the x-axis, and down-regulated genes are shown below the x-axis. Non-expressed genes are shown with empty circles on the x-axis (normalized count < 10 in all samples). **(b,e,h)** The reference sequence and mutated alleles at the 3 CBS hotspots. The mutations in tumors with expression data are underlined (black underline: TCGA tumors, grey underline: SG tumors). **(c,f,i)** The gene expression of *CENPQ* **(c)**, *KCNQ5* **(f)** and *SPG20* **(i)** in normal gastric tissue, and tumors with and without mutations at the corresponding CBS hotspot. *P*-values were adjusted using the Benjamini-Hochberg method.

[Page 16]

Interestingly, the tumor with the highest expression of *CENPQ* was mutated at the highly conserved position 9 of the CTCF motif, while the other two tumors were mutated at position 2 of the CTCF motif. This indicates that different mutations in the same hotspot may have different disruptive potentials. However, a formal evaluation of such effects requires a larger set of tumor samples with both CBS mutations and RNA-seq data available.

Reviewer 3, Comment 1: “The manuscript now reads better as the authors have clarified some issues.”

Reviewer Comment	The manuscript now reads better as the authors have clarified some issues.
Author Response	We would like to thank all reviewers for their appreciation of our work, and for providing constructive suggestions to improve the manuscript.

Reviewer 3, Comment 2: Functional roles of hotspot mutations in the flanking regions of the CTCF motif

Reviewer Comment	They used their statistical pipeline to find CBS hotspots but most of the mutations are outside the CTCF motif so it is less intuitive to think that they are functional. One way to address this is to see if there are motifs for other TFs at positions -5 to 0 in the CBS hotspots or all CBS mutations in Fig 4f. The response, only repeats arguments from the previous version of the manuscript.
Author Response	The reviewer raises an interesting question on the functional consequences of the hotspot mutations in the flanks of the CTCF binding motifs. To examine the possibility that mutations in the flanking regions of CTCF motifs create or disrupt binding motifs of other TFs, we used DeepBind to predict the effect of CBS hotspot flanking mutations on transcription factor binding. We found mutations in CBS flanks of 2 hotspots to create binding motifs for ATF2 and RCOR1. And mutations in CBS flanks of another CBS hotspot to disrupt the binding of SIN3A (Supplementary Table 6). A previous study found that the flanking sequences of weaker CTCF binding sites (those with lower match scores to the CTCF positional weight matrix) are more conserved and could be important for context-specific CTCF binding at these sites (Essien et al., Genome Biol, 2009, doi: 10.1186/gb-2009-10-11-r131). The same observation was also found for REST binding sites (Bruce et al., Genome Res, 2009, doi:10.1101/gr.089086.108). It is therefore possible that the hotspot mutations in the flanking sequences of the CTCF motifs could affect CTCF binding by altering the sequence context of the CBSs. We examined the evolutionary constraints (PhyloP scores based on placental mammals) in the flanking sequences of the mutated CBSs. In general, the 5' flanks of the CBS motif were not conserved, but the 3' flanks of the CBS motif tended to be conserved (Supplementary Fig 5a). Interestingly, at the hotspot upstream of CENPQ, there is a cluster of mutations at the 5' flank of the CTCF motif that coincide with a

	cluster of conserved bases. Therefore, the 5' flank of this hotspot is under evolutionary constraint and is likely important for CTCF function at this site. In addition, we found another CBS hotspot where 9 mutations in the 5' flank of the CBS motif coincided with a highly conserved base (Supplementary Fig. 13). Finally, it is also possible that some of the mutations in the flanking regions of the CTCF motifs are just passenger mutations that arise due to the elevated mutation rates at CBSs. While our model allows identification of individual CBS with overall mutation enrichment, it does not allow us to distinguish between passenger and driver mutations within such a region. In the revised manuscript, we have added the evolutionary conservation analysis in Supplementary Fig. 13, motif creation/disruption analysis in Supplementary Table 6, and added a paragraph in the discussion on the potential roles of the CBS hotspot flank mutations.
Changes to manuscript	[Page 18] Many of the hotspot mutations were located in the 5' flanks of the consensus CTCF motif (Fig. 4f). Previous studies have found increased conservation of the flanking sequences of weaker CTCF and REST binding sites, suggesting that the sequence context is important for TF binding at these sites^{51, 52}. We examined evolutionary conservation⁵³ at the CTCF binding motifs and their flanking sequences. In general, the 5' flanks of the CTCF motifs were not conserved (Supplementary Fig. 13a). However, in the hotspot upstream of CENPQ, the mutation cluster in the 5' flank co-occurred with conserved bases (Supplementary Fig. 13b). In addition, we found another CBS hotspot with 9 5'-flank mutations that coincided with a highly conserved base (Supplementary Fig. 13c). Such hotspot mutations, affecting conserved 5' flanks of CTCF motifs, could disrupt context-specific binding of CTCF. We also examined the possibility that mutations in the flanking regions of CTCF motifs create or disrupt binding motifs of other TFs. We used DeepBind⁵⁴ to predict the binding scores of wildtype and mutated sequences for 472 transcription factors. However, we only found mutations at three CBS sites with predicted change in TF binding (Supplementary Table 6). Lastly, it is also possible that some mutations at CBS flanks are passenger mutations arising due to the overall elevated mutation rates at CBSs. While our model identifies individual CBS regions with overall mutation enrichment, it does not allow us to distinguish between passenger and driver mutations within such regions.

[Supplementary Fig. 13]

Evolutionary conservation of the consensus CTCF motif and flanking sequences. (a) Average PhyloP scores of the CTCF-binding motif and ± 5 flanking bases of all mutated CBSs. (b-c) Two CBS hotspots (b is hotspot upstream of *CENPQ*) where mutations at 5' flanks of CTCF-binding motifs coincide with conserved bases.

[Supplementary Table 6]

DeepBind analysis of hotspot mutations in the flanking regions of CTCF-binding motifs.

Reviewer 3, Comment 3: Presentation of expression analysis

Reviewer Comment	This analysis now has been revised and it turns out that after correction for multiple testing the change in expression for SPG20 is no longer significant. This should be made clear to the reader so in the section “CBS hotspot mutations alter expression of neighboring genes” they should write “We found genes with nominally altered expression for 3 of the 4 hotspots” and in Discussion “Out of the 4 CBS hotspots examined, 3 of them were associated with nominally significant expression changes.....”. In Fig 5 c, f and I they still show the nominal p-values which is not appropriate. They should either change to the values corrected for multiple testing or show both. They still have an argument that the genes change in the same direction in wt tumors as compared to normal tissue so the finding is suggestive but should be presented in a more objective way.
Author Response	The reviewer made a valid point that the FDR corrected P-values should be shown in Fig. 5, and we apologize for this omission in the previous revision. In this revision, we have added the FDR corrected P-values in Fig 5 c, f, and i. We have also edited the manuscript to clarify that 3 out of 4 CBS hotspots were associated with nominally significant expression changes of nearby genes, and 2/4 are associated with significant expression changes after multiple testing correction.
Changes to manuscript	[Abstract] In 3 out of 4 tested CBS hotspots, mutations were nominally associated with expression change of neighboring genes (CENPQ, KCNQ5, SPG20). [Page 16] We found genes with nominally altered expression for 3 of the four hotspots (Fig. 5), two of them remain significant after correcting for multiple testing in each region. [Page 24] Out of the 4 CBS hotspots we examined, 3 of them were associated with nominally significant expression changes of neighboring genes (CENPQ, KCNQ5 and SPG20)

Reviewer 3, Comment 4: Previous validations on CBS mutations

Reviewer Comment	This reviewer also asked if the CBS mutations have been seen before in other studies, regardless of any evidence of positive selection but apparently the authors misunderstood the question. It would be interesting to know if the mutations have been seen before in other tumors and this can be presented in a supplementary table and commented on in the text.
------------------	---

Author Response	We thank the reviewer for clarifying this question. In addition to our existing figure 6 (summarizing the presence of hotspot mutations across 19 TCGA cancer types), we have now also checked if the mutations are listed in COSMIC, Katanein et al. (ref 14) and Umer et al. (ref 39) (Supplementary Table 7). We found that mutations in all 11 CBS hotspots were listed in COSMIC, mutations at 7/11 were listed in supplemental table 4 of Katainen et al., and 4/11 were listed in supplemental table 5 of Umer et al. As suggested by the reviewer, we have commented on this and added a new supplementary table 7 in the revised manuscript.
Changes to manuscript	[Page 20] Similarly, we found that mutations at all CBS hotspots had previously been reported in COSMIC⁵⁶ or other genome-wide studies of gastrointestinal tumors^{14, 39} (Supplementary Table 7). [Supplementary Table 7] Table of CBS hotspot mutations identified in previous genome-wide studies of gastrointestinal tumors and the COSMIC database.

Reviewer 3, Comment 5: Association between SCNA and CBS mutations in the GS subtype

Reviewer Comment	It is still an issue that they find association between CBS mutations and SCNA only in CIN tumors even though they are present also in GS tumors (Fig 4b). I wonder if they could clarify this by repeating the analysis in CIN presented in Fig 7 c, d also in GS tumors.
Author Response	The reviewer made a good suggestion to check the association between CBS mutations and SCNA in the GS subtype. Accordingly, we repeated the analysis of Fig 7b for the GS tumors. Similar to the CIN tumors, the mutated CBSs tended to be closer to SCNA breakpoints compared to the non-mutated CBSs. However this difference was not significant in the GS tumors (Supplementary Fig.14), and the relative difference was greater in CIN (2.17-fold difference in distance to nearest breakpoint) compared to GS (1.58-fold difference) tumors. Interestingly, this may indicate that the coupling of CBS mutations and chromosomal instability is a process that is specific to, or exacerbated, the CIN tumors.

	We note that the alternative analysis in Fig 7c-d cannot be applied to the GS subtype because there are very few genomic windows with high SCNA levels in the GS tumors. In fact, if we bin genomic windows of GS tumors using the same binning thresholds in Fig 7c-d, >94% of the genomic windows are in the first 2 bins.
Changes to manuscript	[Page 22] As the CBS mutation rate was also elevated in GS tumors (Fig. 4b), we investigated if there was a similar association between CBS mutations and SCNA in GS tumors. Although we found that mutated CBSs also tended to be closer to SCNA breakpoints compared to the non-mutated CBSs in GS tumors, the difference was not statistically significant (Supplementary Fig. 14), and the relative difference was greater in CIN (2.17-fold difference in distance to nearest breakpoint) compared to GS (1.58-fold difference) tumors. This may indicate that the coupling of CBS mutations and nearby chromosomal instability is a process that is specific to, or exacerbated in, the CIN tumors. [Supplementary Fig. 14]  Distance to the nearest CNV breakpoint from CBSs at loop boundary and non-boundary CBSs for GS tumors.

Reviewer 3, Comment 6: Add discussion on the limitations of the statistical methods in non-coding driver discovery

Reviewer Comment	Finding non-coding mutations that contribute to cancer is a challenging but interesting area, which the scientific community is getting better at but the methods can still improve, whether statistical or based on wet lab experiments.
---

	It is therefore appropriate to comment on this, perhaps at the end of the paper, with a discussion on the fact that we do not know all biases that exist in the processes of mutation and repair, which gives limitations to the statistical methods. This is also warranted by the fact that the 23 non-CBS hotspots never co-localized with TF-binding regions which makes the biological interpretation less logical.
Author Response	We agree with the reviewer that since we do not know all mutational biases that exist, some of the mutation hotspots identified could be due to unknown mutational biases instead of positive selection. In this manuscript, we tried to correct for as many mutational biases as possible by incorporating epigenetic and sequence context features into the model. We have also examined the hotspots manually to check for broader sequence context biases such as palindromic sequences, and we did not find any. However, it is likely that there are unknown covariates that we have not accounted for, and the unknown covariates limit the accuracy of cancer driver identification methods, potentially leading to both false positive and false negative discoveries. In the revised manuscript, we have added this discussion on the limitations of cancer driver identification methods.
Changes to manuscript	[Page 26] The statistics of cancer driver identification is still limited by our knowledge of the somatic mutation and repair processes. Although our background model corrected for many covariates of the somatic mutation rate, such as epigenetic and sequence context features, false positives and false negatives could still arise from the current model not considering such unknown mutational biases.

REVIEWERS' COMMENTS:

Reviewer #1 (Remarks to the Author):

The revised manuscript by Guo et al. now addresses all remaining questions I had. I welcome the fact that they changed the title to remove the term "positive selection" from it (as also requested by reviewer 2) and they also admit in their rebuttal "While our model allows identification of individual CBS with overall mutation enrichment, it does not allow us to distinguish between passenger and driver mutations." (The only reservation I have is that they still list positive selection among the "manuscript highlights" below the abstract.)

Other than that, I would say it is now up to the reader to decide whether the identified loci are likely to drive gastric cancer (or if follow-up experiments should be done) – the authors have now made all steps in their analysis clear, the analysis is well set up and explained, and in a cutting-edge area of cancer genetics.

Reviewer #2 (Remarks to the Author):

The authors did not address my comments fully, however, this is because of lack of suitable cases.

Reviewer #3 (Remarks to the Author):

The authors have now given relevant answers to all my questions so I am happy for this paper to be published.